# Graft survival of major histocompatibility complex deficient stem cell-derived retinal cells
Masaaki Ishida[1,2], Tomohiro Masuda[1,3,4], Noriko Sakai[1,3], Yoko Nakai-Futatsugi ⓘ [1,3,4] ✉, Hiroyuki Kamao[5], Takashi Shiina ⓘ [6], Masayo Takahashi[1,4,7,8] & Sunao Sugita[1,7,8] ✉

## Abstract

**Background** Gene editing of immunomodulating molecules is a potential transplantation strategy to control immune rejection. As we noticed the successful transplantation of retinal pigment epithelium (RPE) derived from embryonic stem cells of a cynomolgus monkey that accidentally lacked MHC class II (MHC-II) molecules, we hypothesized immune rejection could be evaded by suppressing MHC-II.

**Methods** Gene editing by the Crispr/Cas9 system was performed in induced pluripotent stem cells derived from a cynomolgus monkey (miPSCs) for targeted deletion of the gene coding *class II MHC trans-activator* (*CIITA*). Then the *CIITA*-knocked out miPSCs were differentiated into RPE cells to generate miPSC-derived MHC-II knockout RPE. The MHC-II knockout or wild-type RPEs were transplanted into the eyes of healthy cynomolgus monkeys. All monkeys used in this study were male.

**Results** Here we show when MHC-II knockout RPE are transplanted into monkey eyes, they show suppressed immunogenicity with no infiltration of inflammatory cells, leading to successful engraftment.

**Conclusions** Our results reasonably evidence the efficacy of MHC-II knockout iPSC-RPE transplants for clinical application.

## Plain language summary

Transplantation of healthy cells can be used to treat irreversibly damaged organs. However, a concern is that the transplanted cells will be rejected by the immune system. Generally, the immune system protects our body when unknown materials invade. But this is undesirable during cell transplantation as the transplanted cells are often eliminated by the host's immune cells. We demonstrated in monkeys that deletion of part of the immune system in cells prior to transplantation reduced the amount of immune system activity following transplantation. Using similar strategies in the future could enable cell transplants to be used more successfully in humans, making cell transplantation therapy safer and applicable to a wider number of patients.

To date, there are over fifty registered clinical studies using pluripotent stem cells[1], which include human cell therapy trials for age-related macular degeneration (AMD) or Stargardt's macular dystrophy in several institutes[2]. Our laboratory pioneered in the establishment of retinal pigment epithelium (RPE) cells from patient-derived induced pluripotent stem cells (iPSCs) followed by autologous transplantation of iPSC-derived RPE (iPSC-RPE) for the treatment of neovascular AMD[3]. However, there is a wide range of limitations in the production of autologous iPSC-derived cells, making autologous transplantation difficult to be a standard therapy. In reality, most of the clinical trials use embryonic stem cell (ESC)- or iPSC-derived allogenic transplants with immunosuppressants[4–11]. As most AMD patients are older in age, the administration of immunosuppressive drugs, which causes several adverse events, should be avoided whenever possible.

Although major histocompatibility complex (MHC)-matched transplantation could avoid immune rejection without immunosuppressants[12], this strategy is also barely cost-effective and hardly covers all the patients in the world[13]. Recently, avoiding immune rejection of cell transplants of various organs by artificially controlling immune-related molecules such as MHC and/or co-stimulatory molecules have been reported[13–18].

Unlike retinal cells that express few MHC class II (MHC-II) molecules[19,20], RPE cells, like antigen-presenting cells (APCs), express both MHC class I (MHC-I) and MHC-II molecules, and are highly immunogenic[21,22]. While MHC-I molecules are constitutively expressed, MHC-II molecules are expressed only under inflammatory conditions, for example, if in vitro by IFN-γ administration[18,23–25]. This applies to monkey-iPSC-derived RPE (miPSC-RPE) cells[18,23–25] as well as human and mouse

[1]Laboratory for Retinal Regeneration, Center for Developmental Biology, RIKEN, Kobe, Japan. [2]Department of Ophthalmology, Toyama University, Toyama, Japan. [3]VC Cell Therapy Inc, Kobe, Japan. [4]Ritsumeikan University, Research Organization of Science and Technology, Kusatsu, Japan. [5]Department of Ophthalmology, Kawasaki Medical School, Okayama, Japan. [6]Department of Molecular Life Science, Tokai University, School of Medicine, Kanagawa Isehara, Japan. [7]Kobe City Eye Hospital, Department of Ophthalmology, Kobe, Japan. [8]Vision Care Inc, Kobe, Japan. ✉e-mail: futatsugi@vcct.jp; sugita@vision-care.jp

iPSC-RPE cells. Although immune rejection occurs in allotransplantation of RPE cells in cynomolgus monkeys[26,27] and in human[28], in the present study, we show RPE cells established from ESCs of one of the monkeys (mESC-RPE) was not rejected after allogenic transplantation, which later we found this mESC-RPE had no expressions of MHC-II molecules, and hypothesized graft rejection could be avoided by suppressing MHC-II expression in RPE cells for transplantation. So, we generated MHC-II knockout monkey iPSCs with the Crispr/Cas9 system, and differentiated them into MHC-II knockout miPSC-RPE cells to analyze the immunogenicity and immune rejection of MHC-II unexpressed monkey RPE, both in vivo and in vitro.

This study reasonably evidences the efficacy of MHC-II knockout strategy for RPE transplantation.

## Methods

### Animals

In this study, seven cynomolgus macaques (*Macaca fascicularis*) were used as recipients. A minimal sample size sufficient for statistical analysis was aimed. The monkeys were purchased from Japan Biological Science (Cyn51), Japan SLC Ltd. (HM20), and Eve Bioscience Ltd. (HM-21, 22, 23, 24, and 25). All procedures for animal care and experiments adhered to the ARVO Statement for the Use of Animals in Ophthalmic and Vision Research, and the Use of Laboratory Animals, as well as to the Guidelines of the RIKEN Animal Experiment Committee. The study was approved by the Animal Experiment Committee of the RIKEN Kobe Institute (Approval ID: A2008-02-15, 05/08/2019). To avoid immune rejection due to male-specific histocompatibility antigen coded on the Y-chromosome (H-Y antigen), all the monkeys (both doners and recipients) used in this study were male. Monkeys were accommodated in air- and light- conditioned cages.

### Cell culture and differentiation of monkey ESCs and monkey iPSCs

The monkey ES cell (mESC) line was derived from CMK-6 cynomolgus monkey[29], and the monkey iPS cell (miPSC) line 1121A1 was derived from HT-1 MHC homozygote cynomolgus monkey[26]. The mESC line was maintained on PMEF-CFX (SIGMA) with DMEM medium (SIGMA). The miPSC line was maintained on iMatrix 511 (Nippi Inc) with StemFit AK02N medium (TakaraBio). mESCs and miPSCs were passaged in small clumps after treatment with accutase (Sigma) and replated onto PMEF-CFX every 5 days in average. For induction of RPE cells, mESCs and miPSCs were cultured on gelatin-coated dishes in GMEM (Gibco) supplemented with CKI-7 (3 μM, Sigma), SB431542 (5 μM, Sigma) and Y-27632 (10 μM, Wako)[26,27]. After the appearance of pigment epithelium-like colonies, they were transferred onto laminin-coated cell culture plates in SFRM (DMEM/F12 [7:3] supplemented with B27 (Invitrogen), 2 mM L-glutamine) supplemented with 10 ng/ml bFGF and SB431542 (0.5 μM). The medium was changed to DMEM/F12 medium with B27 supplement (Gibco). The differentiation medium was changed every 2–3 days.

### Generation of *CIITA* knockout miPSCs

Frozen stocks of miPSC were seeded on 6-well plates coated with iMatrix511 at $2 \times 10^5$ cells per well. They were cultured with StemFit AK02N iPSC maintenance media in the presence of Y27632 (10 μM) for the first 24 hr. Transfection was performed on day 1 with 2.5 μl of lipofectamine 3000 transfection reagent (Invitrogen) and 2.5 μg of plasmid (px459v2) that expresses the sgRNA of *CIITA* or *RFXANK*. The cells were dissociated into single cells and seeded on iMatrix-coated 6-well plates at $6 \times 10^3$ cells per well at day-2. The number of cells and the concentrations of lipofectamine and plasmid DNA were optimized in advance. The medium was changed every day. Y27632 was added for the first 24 hr (10 μM). On day-5 or −6, colonies were picked up (96 colonies per one trial). The half of each colony was passaged and the other half was subjected to genotyping. Candidate colonies were continued with culture and passages. After three trials of this process, the knockout cell lines described in Supplementary Fig. 4b-c were obtained. For genotyping, the primers shown in Supplementary Table 1 were used.

### Subretinal transplantation in animal models

All surgeries were performed in a surgery room for monkeys. All monkeys were anesthetized with a mixture of ketamine and xylazine. The pupils were dilated with 0.5% tropicamide and 0.5% phenylephrine hydrochloride. For transplantation of ESC/iPSC-RPE cells, the standard three-port pars plana vitrectomy was performed under this anesthesia. After core vitrectomy, triamcinolone-assisted posterior vitreous detachment was carried out, and the vitreous was removed as much as possible (Constellation®, Alcon, Geneva, Switzerland). A localized retinal detachment was created by intraocular irrigating solution (BSS Plus; Alcon) using PolyTip® Cannulas (Cat #3219; MedOne Surgical Inc) and VFC system (Constellation®, Alcon, Geneva, Switzerland). ESC-RPE sheets were transplanted in the same way as described previously[26]. ESC-RPE cells and iPSC-RPE cells (single-cell suspensions: $1–2 \times 10^5$ cells/eye) were transplanted into the subretinal space at one injection with the PolyTip® Cannulas connected to a 1 mL MicroDose injection device (Cat #3275; MedOne Surgical, Inc.) filled with single-cell suspension. In order to trace the grafts after transplantation, RPE cells were labeled with PKH fluorescent dyes (Sigma-Aldrich: Cat #PKH26GL) before transplantations. Laser photocoagulation was performed one month before operation. Cyn51 monkey had total 18 shots (9 spots around the fovea and 9 spots at lower macular) under the following conditions: spot size, 80μm; duration, 100 msec; power, 350-450 mW; wavelength, red laser.

The graft was monitored and we observed the symptom of immune rejection using color fundus photos, fluorescein angiography (FA), auto-fluorescein phots (AF) and spectral domain optical coherence tomography (OCT) after surgery at each time point (1 W, 2 W, 4 W, 8 W, 3 M, 4 M, 5 M, and 6 M). The OCT scan (RS-3000, NIDEK) was performed to identify the survival and engraftment of grafted cells and to detect immune rejection such as retinal edema or thinning of the retinal layer. We also followed the engrafted grafts as much as possible. FA (RetCam II or RetCam III or CX-1) was conducted to find leakage of fluorescence resulted from immune rejection.

### MHC typing for animal models

The Mafa class I and II haplotypes of the cynomolgus macaques were determined by the Sanger sequencing method and high-resolution pyrosequencing[30]. Results of MHC genotyping are noted in Supplementary Data 1.

### Immunohistochemistry

Monkey eyes enucleated at 6 months after RPE cell/sheet transplantation were fixed in formaldehyde (Super Fix, Kurabo) and embedded in paraffin (Sigma-Aldrich). Paraffin sections were sliced into 10-μm-thick sections in a series of five sequential slides by using an autoslide preparation system (Kurabo). Hematoxylin and eosin (H&E) staining were performed as described previously[27,31]. For immunohistochemistry (IHC), sections were blocked with 5% goat serum in PBS for 1 hr at room temperature, followed by overnight incubation with the following primary antibodies at 4 °C: anti-ionized calcium-binding adapter molecule 1 (Iba1), anti-CD3, anti-MHC class II (MHC-II) and anti-CD4 (Supplementary Table 2). After rinsing with Tween 20 in PBS three times, sections were incubated with appropriate secondary antibodies (Supplementary Table 2) for 1 hr at room temperature and counterstained with DAPI (×1,000; Life Technologies). Images were acquired with a confocal microscope (LSM700, Zeiss). Staining and microscopy were performed by two technical staffs who were not informed of the study design.

### Lymphocyte-graft immune reaction test (LGIR) test

The LGIR test was developed as an in vitro assay for the evaluation of immune rejection and for the selection of efficacious medications[22,24,32], which has been used in our clinical trials[1,12]. Blood sample was collected from each monkey (Cyn51, HM-20, 21, 22, 23, 24, 25, Okieso, Kijihata, S2-6, and TLHM-6). mESC/miPSC-RPE cells that were treated with 20 Gy radiation ($1 \times 10^4$/well) were co-cultured with PBMCs ($1 \times 10^6$/well) isolated from the blood samples for 5 days. As a positive control of immune activation,

EBV-transformed B cells were used instead of RPE cells for co-culture with PBMCs. After co-culture with RPE or EVB-B cells, the PBMC samples were collected and stained with the antibodies listed in Supplementary Table 2. For intracellular staining for Ki-67, the cells were fixed and permeabilized (BioLegend). After staining, the cells were analyzed with FACS CantoTM II controlled by FACSDiva v6.0 (BD). The population was determined by staining with lymphocyte markers CD4, CD8, CD11b, and NKG2A, and analyzing 1 million cells for each sample, which showed sufficient purity and abundance. The FSC/SSC gates were determined by our preliminary study to include the population positive for lymphocyte markers (e.g. CD4, CD8, CD11b, or NKG2A). For Ki-67 staining, the gate was extended to higher FSC/SSC to include proliferating cells. For data analysis, FlowJo software (version 9.3.1) was used. Sample preparation and analysis were performed by a technical staff who was not informed of the study design.

## Quantitative RT-PCR

RPE-specific markers in two iPS-RPE cell lines (CIITA$^{+/+}$ or CIITA$^{-/-}$ iPSC-RPE cells) were examined by quantitative RT-PCR (qRT-PCR). The mRNA expressions for PEDF, tyrosinase, TGF-beta2, RPE65, and Pax6 were evaluated. Total RNA was isolated from the iPS-RPE cells. After cDNA synthesis, the expressions of the above molecules and β-actin in experimental triplicates were analyzed by qRT-PCR with a LightCycler 480 instrument and qPCR Mastermix and Universal ProbeLibrary assays (Roche Diagnostics, Mannheim, Germany) in the following condition[33,34]: Denaturation at 95 °C for 10 min, followed by 45 cycles of denaturation at 95 °C for 10 sec, annealing at 60 °C for 30 sec, and an extension at 72 °C for 1 sec. Relative mRNA expression was calculated with Relative Quantification software (Roche Diagnostics) by using an efficiency-corrected algorithm with standard curves and reference gene normalization against that of β-actin (delta delta cycle threshold [$\Delta\Delta C_t$]). Results are indicated by the relative expression of the molecules ($\Delta\Delta C_t$: control cells = 1). The primers and probes are as follows: *PEDF*, L: gtgtggagctgcagcgtat, R: tccaatgcagaggagtagca, probe #57; *tyrosinase*, L: gctgccaatttcagctttaga, R: ccgctatcccagtaagtgga, probe #47; *TGF-beta2*, L: ccaaagggtacaatgccaac, R: cagatgcttctggatttatggtatt, probe #67; *RPE65*, L: caatgggtttctgattgtgga, R: ccagttctcacgtaaattggcta, probe #84; Pax6, L: ggcacacacacattaacacactt, R: ggtgtgtgagagcaattctcag, probe #9; *β-actin*, L: ccaccgcgagaagatga, R: ccagaggcgtacagggatag, probe #64.

## Flow cytometry

Expressions of MHC-I and MHC-II on RPE cells were examined by FACS analysis. Before staining, RPE cells were incubated with an Fc block (Miltenyi Biotec) at 4 °C for 15 min. mESC-RPE and miPSC-RPE cells co-cultured with recombinant IFN-γ (100 ng/mL) for 48 hr were also prepared. The cells were stained with anti-MHC-I antibody, anti-MHC-II antibody, or isotype control (mouse IgG) at 4 °C for 30 min. Expressions of co-stimulatory molecules CD40, CD80 (B7-1), CD86 (B7-2), B7-H1 (PD-L1) and CD275 (ICOS-L/B7-H2) were also examined. Expressions of RPE-specific markers such as RPE65, bestrophin, MerTk, MiTF, Pax6, ZO-1, PEDF, tyrosinase, and OCT3/4 (iPSC-marker) were evaluated by FACS analysis as well. Anti-rabbit IgG or anti-mouse IgG (isotype controls) were used to define the baseline of the staining. All information of the antibodies are in Supplementary Table 2.

To confirm phagocytic function of the shed photoreceptor rod outer segments (ROS), we cultured mESC-RPE cells in the presence of fluorescein isothiocyanate (FITC)-labeled porcine ROS for 5 hr at 37 °C, and phagocytosis was evaluated by FACS[33,34].

FACS was performed with FACS CantoTM II controlled by FACSDiva v6.0 (BD). The population was determined by analyzing 1 million cells for each sample. The FSC/SSC gates were determined by our preliminary study to include the population positive for RPE markers (e.g. RPE65). For data analysis, FlowJo software (version 9.3.1) was used.

Experiments were performed by a technical staff who was not informed of the study design.

## Western blots

Cells (RPE or PBMCs) were lysed and boiled in a Sample Buffer, Laemmli 2x Concentrate (SIGMA-ALDRICH). Proteins were separated by 5–20% gradient SDS–PAGE gel (SuperSep Ace, FUJIFILM) and then transferred to PVDF membrane (BIO-RAD). Membranes were blocked with Blocking One (Nacalai tesque) in for 1 h and subsequently exposed to primary antibodies overnight, and then to secondary antibodies for 1 h. The proteins were detected with Chemi-Lumi One Super (Nacalai tesque). αTubulin was used as a loading control. The following primary antibodies were used: anti-CIITA rabbit antibody (GENETEX, GTX129022 1:100), anti-MHC class II mouse antibody (abeomics, 36-1236 1:500) and anti-αTubulin mouse antibody (SIGMA-ALDRICH, DM1A, 1:5000).

## Statistics and reproducibility

For statistical analyses, Microsoft Excel 2019 was used. There were no exclusions of data points. Randomization was not applied and confounders were not controlled in this study.

The exact $P$ values for Figs. 5c and 7c are in the Figure legend.

## Reporting summary

Further information on research design is available in the Nature Portfolio Reporting Summary linked to this article.

## Results

### Ophthalmological tests after transplantation of mESC-RPE cells

First, we examined whether mESC-RPE cells are applicable to retinal transplantation in monkey in vivo models. We established mESC-RPE cells from ESCs derived from one of the experimental cynomolgus monkeys (cynomolgus macaques: *Macaca fascicularis*) CMK6, and transplanted into other monkeys (monkey IDs: Cyn51, HM-20, and HM-21) as allografts. The experimental design is described in Fig. 1a. Immunosuppressants were not used for the transplantation. The mESC-RPE cells were qualified as RPE-transplants by their polygonal and hexagonal morphologies, and sufficient degree of pigmentation (Fig. 1b) as well as other RPE-specific functions (Supplementary Fig. 1).

The expressions of MHC-I and -II molecules were assessed by stimulating the mESC-RPE cells with recombinant IFN-γ in vitro. MHC-I was constitutively expressed as expected. However, even in the presence of IFN-γ, the mESC-RPE cells derived from CMK6 failed to express MHC-II on the cell surface (Fig. 1c), which was in contrast to our previous report showing human ESC-derived RPE cells, as well as primary fetal RPE cells, exposed to IFN-γ expressed MHC-II[24]. The genotyping of MHC-I and MHC-II genes by pyrosequencing[30] showed MHC-II gene (Mafa class II) was undetectable (Supplementary Data 1), suggesting *MHC-II* was not expressed at mRNA level in mESC-RPE cells derived from CMK6.

When these *MHC-II*-deficient mESC-RPE cells were transplanted in the form of a cell sheet into the subretinal space of the eyes of a normal monkey (Cyn51) (Fig. 1d) without immunosuppressants, there was no rejection at least for 2 years, shown by fundus photographs and optical coherence tomography (OCT) with pigmented RPE-sheet surviving without symptoms of immune rejection (Fig. 1e). At the final OCT examination (24-month), the grafted RPE-sheet seemed to be integrated with host RPE, which was in contrast to our previous study that showed OCT images of full rejection after money allogeneic RPE transplantation[31], indicating the transplanted RPE cell sheet survived for a long period without rejection.

### Immunohistochemical staining of retinal sections after transplantation of ESC-RPE allografts

Although our first clinical trial for iPSC-RPE transplantation was realized in the form of RPE-sheet transplantation[3], we are currently moving to a less invasive form. Thus, instead of mESC-RPE-sheet transplantation described above (Fig. 1d, e), we next transplanted the same CMK6-derived mESC-RPE into other 2 monkeys (HM-20 and HM-21) in a form of cell suspension (Fig. 2a) without immunosuppressants. As in Cyn51 (Fig. 1e), immune rejection against *MHC-II*-deficient allogeneic mESC-RPE cells was not

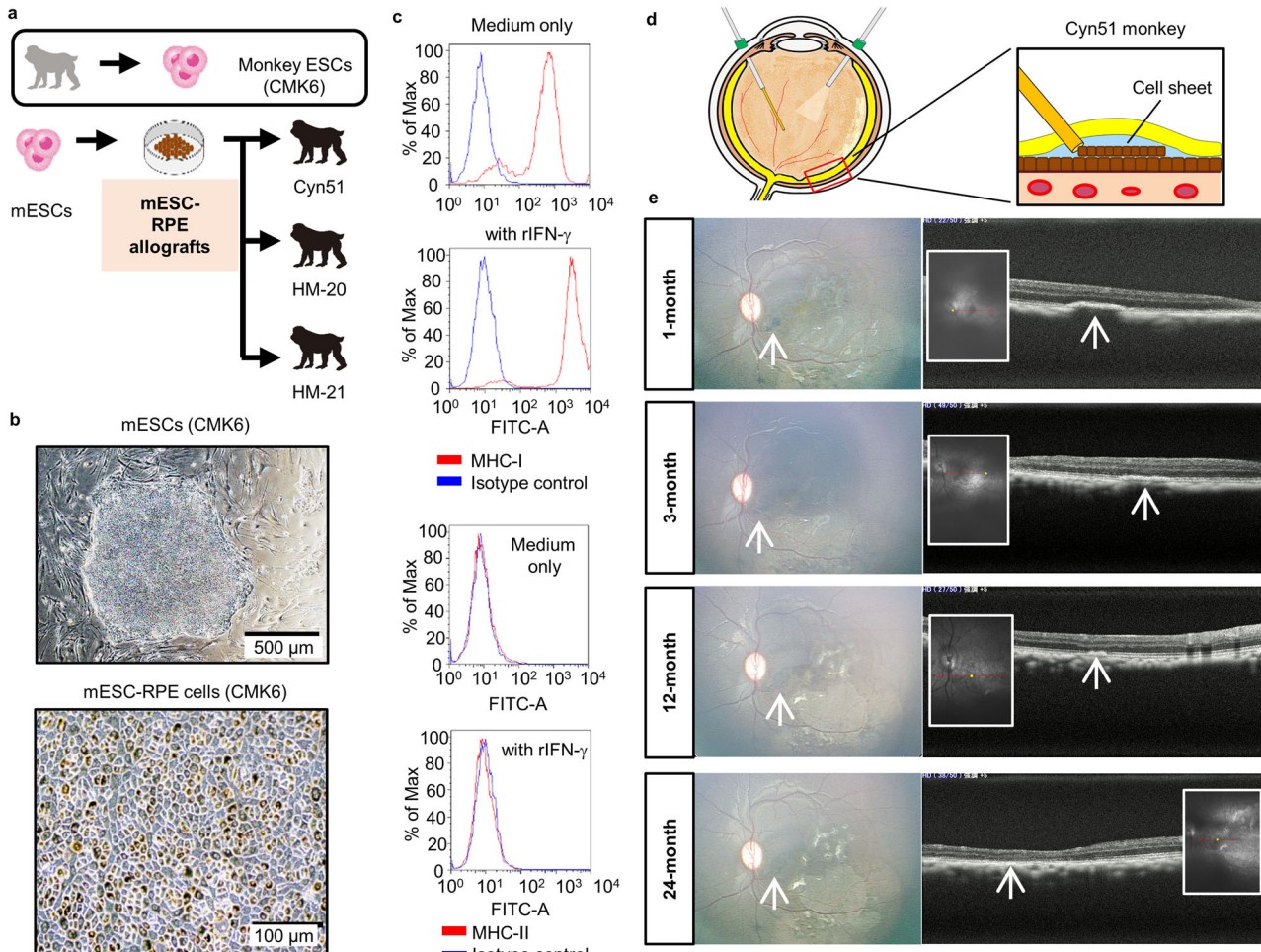

**Fig. 1 | Characterization of CMK6-derived ESC-RPE cells and the transplantation without immunosuppressive drugs. a** Experimental design for the transplantation of RPE cells established from monkey CMK6-derived ESCs. Totally 3 normal adult monkeys (Cyn51, HM-20, HM-21) were transplanted. **b** RPE cells (lower panel) differentiated from CMK6-derived ESCs (upper panel) showed normal appearance with hexagonal morphology and pigmentation. Scale bars, upper: 500 μm, lower: 100 μm. **c** Expressions of MHC class I and class II molecules in CMK6-derived mESC-RPE. mESC-RPE cells were stimulated with recombinant

IFN-γ (100 ng/mL) for 48 h. CMK6-derived mESC-RPE cells did not express MHC class II. **d** Schema of vitrectomy for mESC (CMK6)-RPE cell sheet transplantation into the subretinal space of Cyn51 monkey. **e** Cyn51 who received mESC (CMK6)-RPE cell sheet did not show rejection at least for 2 years. Pigmented sheet (white arrows) was detected in color fundus (left) and OCT (right) without symptoms of immune rejection at all observation time points (1, 2, 12, and 24 months). At 24 months, sheet integration with the host RPE was observed by OCT.

observed by basic ophthalmologic examinations during the planned observation period (6 months). Further examination by color pictures, fluorescein angiography, and OCT at 1, 3, and 6 months, showed sheet-like grafted RPE cells with no signs of rejection (Fig. 2b and Supplementary Fig. 2). In Hematoxylin and Eosin (H&E) staining, grafts were identified by their characteristic appearance: generally darker than host RPEs, tended to form not a monolayer but thick multilayer, sometimes overlaying the host RPE and sometimes migrating beneath the host RPE, eventually integrate and become sheet-like. As shown in Fig. 2c, survival of transplanted mESC-RPE cells was detected in both eyes of HM-20 and −21 monkeys. By immunohistochemistry (IHC) of the retinal sections (Fig. 2d), CD3[+] T cells, CD20[+] B cells, and MHC-II[+] APC were not detected, although infiltration of a small number of Iba1[+] cells (=microglia/macrophages) at the grafted area was observed (Fig. 2d). This was in contrast to our previous study that showed full rejection after money allogeneic RPE transplantation[31] (Supplementary Fig. 3).

### Generation of RPE graft cells from MHC class II knockout miPSCs

The transplanted RPE cells established from mESCs of CMK6 that lacked MHC-II survived for a long period without immune rejection. Therefore,

we next examined whether RPE cells differentiated from MHC-II knockout iPSCs would have low immunogenicity and exhibit graft survival after transplantation. The experimental design is described in Fig. 3a. An iPSC line, 1121A1, derived from HT-1 MHC homozygote monkey was used[27,35]. First, the gene coding class II MHC transactivator (*CIITA*), a well-known transactivator that activates MHC class II genes[36], was knocked out from 1121A1 using the CRISPR/Cas9 genome editing system, by which we obtained two *CIITA*[-/-] iPSC lines (cell #1, Supplementary Fig. 4b; cell #2, Supplementary Fig. 4c). Then the *CIITA*[-/-] iPSCs were differentiated into RPE cells to eventually obtain *CIITA*[-/-] miPSC-RPE cells (Fig. 3b). As a control, *CIITA*[+/+] miPSC-RPE cells (wild-type) were also prepared. These RPE cells showed hexagonal morphology with rich pigmentation (Fig. 3c). After quality control tests[3,24] (Supplementary Fig. 5), the *CIITA*[-/-] miPSC-RPE cells were considered for transplantation.

CIITA expression was also examined by western blot. *CIITA*[+/+] control miPSC-RPE cells and peripheral blood mononuclear cells (PBMC) of a monkey clearly expressed CIITA proteins with or without IFN-γ, whereas it was not detected in *CIITA*[-/-] miPSC-RPE cells (Fig. 3d and Supplementary Fig. 6). Accordingly, while MHC-II was expressed on the cell surface of *CIITA*[+/+] miPSC-RPE in response to IFN-γ (Supplementary Fig. 7), which was consistent with our previous report[22], *CIITA*[-/-] miPSC-RPE cells did not

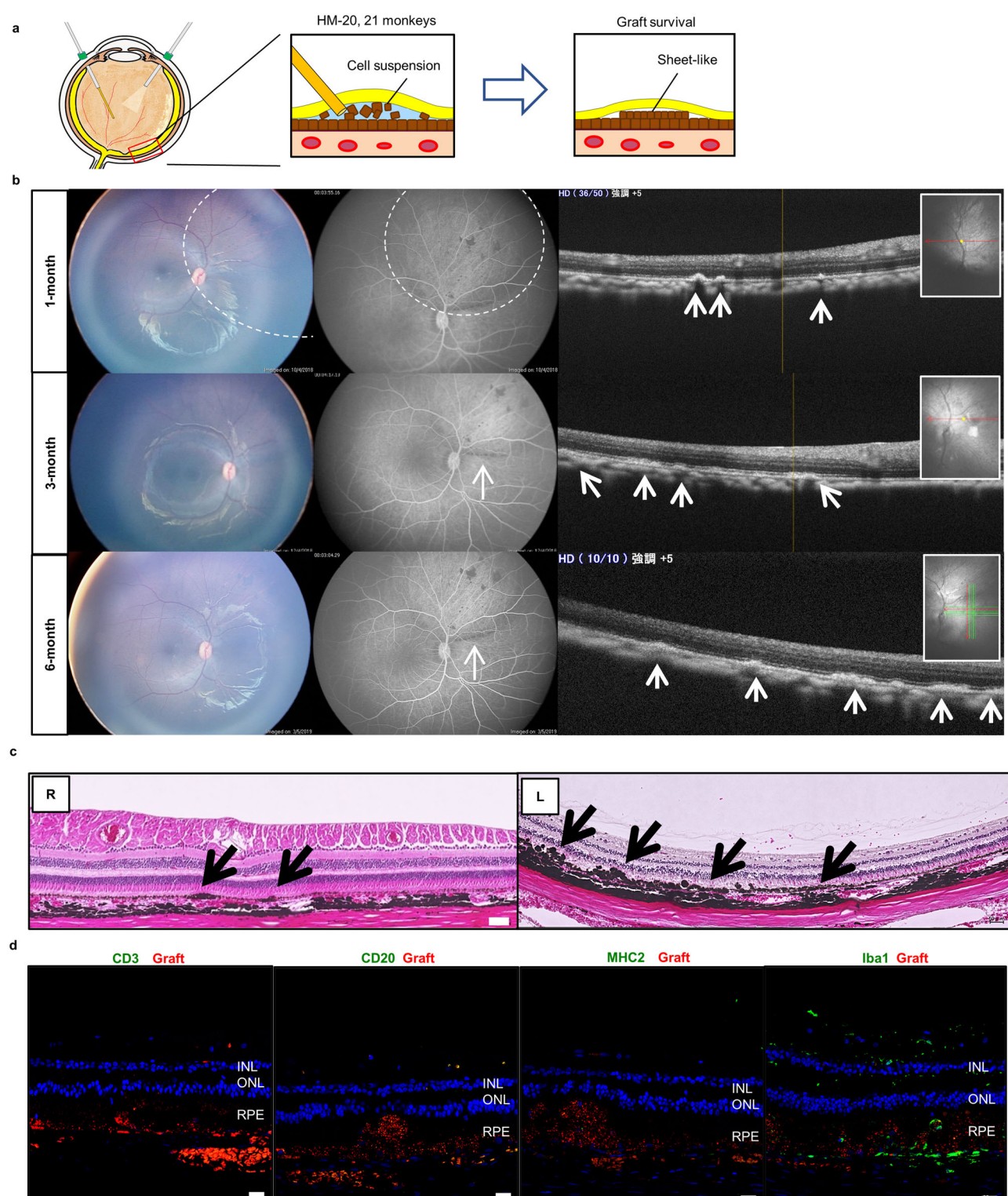

express MHC-II molecules even when stimulated by IFN-γ (Fig. 3e). These results indicated that we successfully generated MHC-II knockout RPE lines qualified for transplantation.

## Transplantation of MHC-II knockout miPSC-RPE cells and ophthalmological tests

The MHC types of recipient monkeys were identified before transplantation, and all cases were allogeneic against the miPSC-RPE graft cells (Supplementary Data 1). The miPSC-RPE cells were transplanted into 4 monkeys (HM-22, HM-23, HM-24, and HM-25): HM-22 received wild-type *CIITA*<sup>+/+</sup> miPSC-RPE cell transplantation, and HM-23, HM-24 and HM-25 received *CIITA*<sup>-/-</sup> miPSC-RPE cell transplantation (Fig. 3a). As a result, monkey HM-22 with wild-type *CIITA*<sup>+/+</sup> allograft transplantation showed no signs of immune rejection by color funds and fundus auto-fluorescent (FAF), but by OCT observation we suspected RPE cell graft-related rejection with inflammatory cell infiltration (Fig. 4a). In contrast,

**Fig. 2 | Transplantation of CMK6-derived ESC-RPE cells in a form of cell suspension without immunosuppressive drugs. a** Schema of the surgery for mESC (CMK6)-RPE cell suspension transplantation into the subretinal space of HM-20, −21 monkeys. If the transplanted cells survive, they should become a sheet-like graft. **b** HM-21 monkey transplanted with mESC-RPE cell suspension was evaluated by color picture (left), fluorescein angiography (middle), and OCT (right) at 1, 3, and 6 months after transplantation. At 6 months, many sheet-like grafted RPE cells were observed by OCT, without any signs of rejection. The left insert of the OCT is a navigation window that indicates the scanning position: OCT image was obtained by scanning along the red line; yellow dot (which is sometimes in a hidden mode) corresponds to the yellow line in the OCT image; when continuous scanning mode is used (usually when the grafted area is not clear), green and red lines indicate the series of scanning position, of which the red line indicates the layer of the OCT image

shown; whether the shown OCT image is the vertical or horizontal red line-layer is indicated by the yellow dot (although the yellow dot is sometimes in a hidden mode). Dashed line circle: retinal bleb made for the injection, white arrows: mESC-RPE grafts. **c** H&E staining of HM-21 monkey 6 months after transplantation. Transplanted mESC-RPE cells were detected in the subretinal space of both eyes. There were no findings of inflammation. R: the right eye. L: the left eye. Arrows: mESC-RPE grafts. Scale bars: 50 μm. **d** IHC of the retinal section of HM-21 monkey. PKH (red) pre-labeling of the grafts was faint after 6 months of transplantation. CD3+ T cells (leftmost panel, green), CD20+ B cells (second left panel, green), and MHC-II+ cells (third left panel, green) were not detected. Infiltration of Iba1+ microglia/macrophages (rightmost panel, green) were detected around the grafted area. DAPI (blue) was used for counterstaining. INL inner nuclear layer, ONL outer nuclear layer, RPE retinal pigment epithelium. Scale bars: 20 μm.

monkey HM-23 who received *CIITA*[−/−] miPSC-RPE (MHC-II knockout) transplantation showed the existence of grafted RPE cells by color funds, FAF, and OCT without immune rejection for 6 months (Fig. 4b). The transplanted cells were engrafted in the eye and became sheet-like. We obtained similar results from other monkeys (HM-24, −25) transplanted with *CIITA*[−/−] miPSC-RPE cells (Supplementary Fig. 8 and Supplementary Fig. 9). In monkey HM-25, *CIITA*[−/−] miPSC-RPE cells were transplanted with Rho-associated protein kinase inhibitor to assess the immune response in a condition that should give more stable engraftment[33].

The results from *CIITA*[+/+] iPSC-RPE cells transplanted into two eyes (HM-22) and *CIITA*[−/−] iPSC-RPE cells transplanted into six eyes (HM-23, −24, −25) indicated transplantation with *CIITA*[−/−] iPSC-RPE cells could be efficient for graft survival without any findings of rejection by general ophthalmic examinations.

## Histochemical findings of MHC-class II knockout miPSC-RPE cell transplantation

Immunohistochemistry of HM-22 monkey transplanted with control *CIITA*[+/+] iPSC-RPE cells showed many CD3+ T cells and CD4+ helper T cells around the grafted RPE cells (Fig. 5a). MHC-II expressing APCs and Iba1+ cells (ameboid-type microglia) accumulated around the grafted cells and in the choroid of the eye as well (Fig. 5a). On the other hand, HM-23 monkey with MHC-II knockout transplantation showed no symptoms of immune rejection. By H&E staining, we are able to find engraftment of the grafted cells (Supplementary Fig. 10b), which was spread to cover a wide area when observed by OCT. By IHC evaluation of the retina transplanted with MHC-II knockout-RPE cells (Fig. 5b), there were almost no infiltration of inflammatory cells such as T cells (CD3+, CD4+), and some microglia or macrophages (MHC2+, Iba1+) were found only in the choroid, which was a condition close to normal eyes (Supplementary Fig. 3a). There were no other inflammatory signs throughout the retinal sections. Similar results were obtained from IHC of other monkeys, HM-24 (Supplementary Fig. 8) and HM-25 (Supplementary Fig. 9), who were transplanted with *CIITA*[−/−] miPSC-RPE cell suspension. Quantification of infiltrated inflammatory cells confirmed less immune response against *CIITA*[−/−] miPSC-RPE transplantation (Fig. 5c and Supplementary Fig. 11).

Taken together, IHC evaluation showed less signs of rejection in eyes transplanted with MHC-II knockout-RPE cells compared to those transplanted with wild-type RPE cells.

## Expression of MHC class II on host RPE layer and grafted RPE cells after transplantation

Next we examined whether the grafted RPE cells express MHC-II molecules after transplantation. We also checked the expressions of MHC-II molecules on the host RPE layer (recipient RPE). For the evaluation, we examined several retinal sections from 4 monkeys: retina that showed rejection after transplantation with 1121A1 wild-type RPE (the right eye of S2-4 monkey that received transplantation in our previous study[27]), retina that showed rejection after transplantation with 46a heterozygote RPE (K-177 monkey that received transplantation in our previous study[27]), retina transplanted

with MHC-II knockout 1121A1RPE without signs of rejection (HM-23 monkey), and normal retina (the left eye of S2-4). As expected, normal retina including the RPE cell layer (host RPE) did not express MHC-II molecules (Fig. 6a). In contrast, the retina that showed rejection after transplantation with wild-type RPE cells expressed MHC-II on the host RPE layer, and inflammatory cells such as Iba1+ retinal microglia and choroidal macrophages were also observed (Fig. 6b). Interestingly, MHC-II was also expressed on blood endothelial cells in the choroid. As shown in Fig. 6c, retina that showed rejection with transplantation of wild-type heterozygote RPE cells expressed MHC-II on the grafted RPE cells, as well as the host RPE layer, and partial destruction of retinal structure was seen in the eye. On the other hand, the retina transplanted with MHC-II knockout RPE (*CIITA*[−/−]) did not express MHC-II on host RPE, grafted RPE cells, inflammatory cells, and blood endothelial cells even though the grafts were allogenic and immunosuppressive drugs were not used (Fig. 6d). These results suggested that rejection accompanied by inflammation (probably the effect of inflammatory cytokines such as IFN-γ) enhanced the expression of MHC-II on both host and graft RPE cells, causing exacerbation of inflammation. By MHC-II deficient RPE cell transplantation, any cells/tissues in the retina were not stained with anti-MHC-II antibody. They were close to normal retina.

Taken together, when MHC-II knockout grafts were transplanted, inflammatory cells did not infiltrate into the grafts, and the grafts were survivable in the retina.

## Immunological blood tests against MHC-class II knockout iPSC-RPE cells

Given that MHC-II knockout was efficient to avoid rejection, we next examined whether the MHC-II knockout cells were indeed hypo-immunogenic. For detection of immune cell clones that responded to the grafted RPEs after transplantation, we performed the lymphocytes-graft cells immune reaction (LGIR) test, which is an in vitro assay with a co-culture of graft cells and blood lymphocytes of the recipient[24,32] (Supplementary Fig. 12a). Due to the immunosuppressive nature of RPE cells[34], co-culture of PBMC with RPE cells often gives less immune activity. However, when PBMCs isolated from HM-22, the recipient monkey of *CIITA*[+/+] miPSC-RPE (wild-type) transplantation, who had shown slight signs of rejection (Fig. 4a), were co-cultured with *CIITA*[+/+] miPSC-RPE (wild-type), there was an increase in proliferative Ki-67+ population of CD8+ cytotoxic T cells, CD11b+ monocytes/macrophages, and NKG2A+ NK cells in response to the graft RPE compared to the condition without the graft (PBMCs only) (Fig. 7a), indicating immune cell clones responsive to *CIITA*[+/+] miPSC-RPE had indeed existed in HM-22 at the time when rejection occurred against *CIITA*[+/+] miPSC-RPE transplantation. In contrast, the same PBMCs did not respond to *CIITA*[−/−] miPSC-RPE (Fig. 7a), indicating the low immunogenicity of the MHC-II knockout RPEs. On the other hand, PBMCs isolated from HM-24 who had received MHC-II knockout transplantation and had not shown immune rejection, did not respond to either *CIITA*[+/+] or *CIITA*[−/−] miPSC-RPE cells by the proliferation of all tested immune cells (CD4+, CD8+, CD11b+, and NKG2A+ cells)

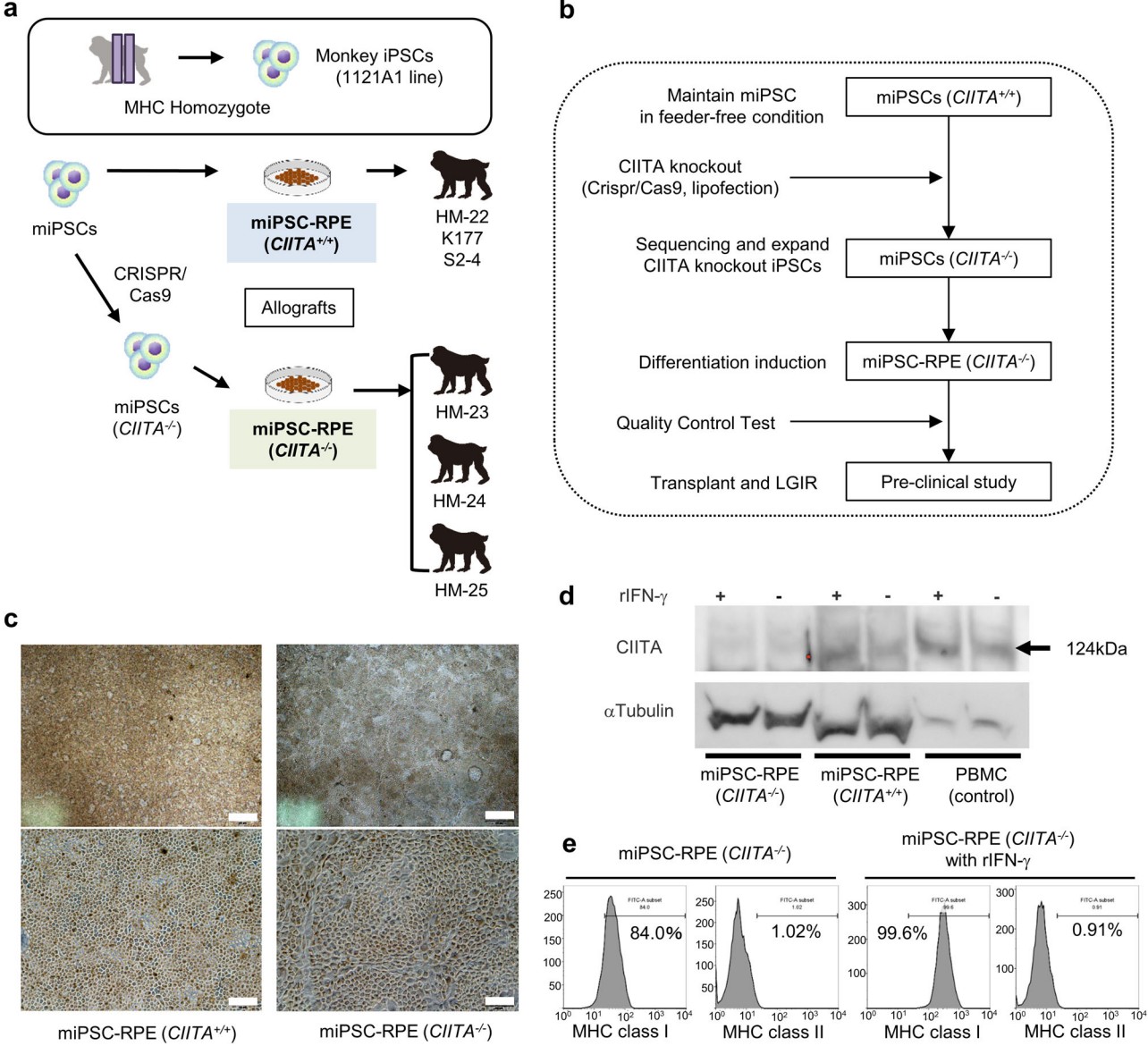

**Fig. 3 | Generation and characterization of *CIITA* knockout monkey iPSCs and RPE cells. a** Experimental design for transplantation of *CIITA*+/+ and *CIITA*-/- miPSC-RPE into 4 healthy monkeys (HM-22 received *CIITA*+/+ miPSC-RPE; HM-23, HM-24, HM-25 received *CIITA*-/- miPSC-RPE). Both miPSC-RPE lines were generated from 1121A1 monkey iPSC line (MHC homozygous). Samples from two other healthy monkeys (S2-4 and K177) who received transplantation of wild-type miPSC-RPE cells (derived from 1121A1 and 46a iPSC lines, respectively) in our previous study[27] were also used in this study. **b** The schema of the *CIITA* knockout production. **c** *CIITA*-/- miPSC-RPE cells (right) show polygonal and hexagonal morphologies as well as the control *CIITA*+/+ miPSC-RPE cells (left). Scale bars: 500 μm (upper) and 100 μm (lower). **d** Western blots for CIITA expression in *CIITA*-/- miPSC-RPE with or without rIFN-γ pretreatment: lane 1 and 2, *CIITA*-/- miPSC-RPE cells; lane 3 and 4, wild type control (*CIITA*+/+); lane 5 and 6, positive control (PBMC from a monkey). Full-length images of the western blots are in Supplementary Fig. 6. **e** Expressions of MHC class I and II on the cell surfaces of *CIITA*-/- miPSC-RPE with or without rIFN-γ stimulation were analyzed by flow-cytometry. X-axis: the FITC-intensity of the cells stained with FITC-conjugated anti-human HLA-class I or anti-human HLA-class II. Y-axis: frequency.

(Fig. 7b), indicating immune cell clones responsive to *CIITA*+/+ or *CIITA*-/- miPSC-RPE cells did not exist in HM-24, which confirmed the low immunogenicity of the *CIITA*-/- miPSC-RPE transplant. We also obtained similar data with PBMCs of monkey HM-20 that was transplanted with MHC-II-deficient mESC-RPE cells (Supplementary Fig. 12b).

While HM-22 showed only a slight sign of rejection (Fig. 4a), which may explain the low immune response of CD4+ T cells against the graft (Fig. 7a), when PBMCs isolated from TLHM6 who had shown severe immune rejection after RPE cell transplantation[31] were subjected to the LGIR test, CD4+ T cells that responded to *CIITA*+/+ miPSC-RPE were detected (Fig. 7c). However, these CD4+ T cells failed to respond the MHC-II knockout *CIITA*-/- miPSC-RPE cells (Fig. 7c), indicating even for the immune cell clones that had become highly active after *CIITA*+/+ miPSC-

RPE transplantation, MHC-II-knocked out *CIITA*-/- miPSC-RPE cells were not immunogenic.

Taken together, by the LGIR test, immune cells responsive to the transplant were not detected in the recipient monkey of MHC-II-knockout transplant, and also low immunogenicity of MHC-II-knockout cells was proven in vitro by showing significantly low response even by highly active immune cells against *CIITA*+/+ miPSC-RPE graft, suggesting the importance of MHC-II molecules expressed on RPE cells for the recognition by immune cells.

## Discussion

In the present study, we demonstrated that deficiency or knockout of MHC-II in transplant RPE cells were efficient in avoiding immune rejection and

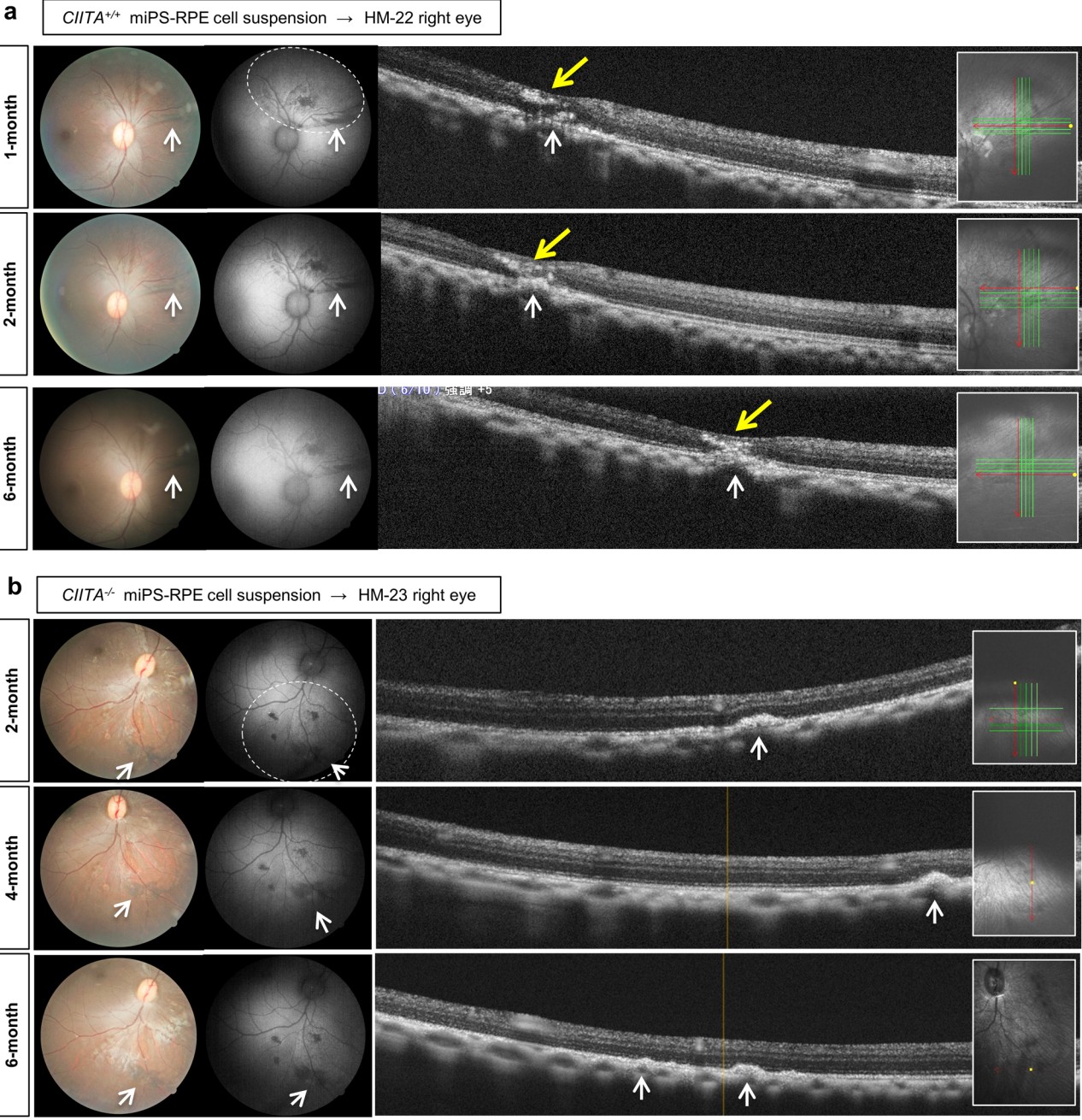

**Fig. 4 | Results of *CIITA*$^{+/+}$ or *CIITA*$^{-/-}$ monkey iPSC-RPE cell transplantation.**
**a, b** Monkey HM-22 transplanted with *CIITA*$^{+/+}$ wild type miPSC-RPE cells (**a**) and monkey HM-23 transplanted with *CIITA*$^{-/-}$ MHC-class II knockout miPSC-RPE cells (**b**) were examined by color funds photograph (left), FAF (middle) and OCT (right) at 1, 3, and 6 months after transplantation. The left insert of the OCT is a navigation window that indicates the scanning position: OCT image was obtained by scanning along the red line; yellow dot (which is sometimes in a hidden mode) corresponds to the yellow line in the OCT image; when continuous scanning mode is used (usually when the grafted area is not clear), green and red lines indicate the series of scanning position, of which the red line indicates the layer of the OCT image shown; whether the shown OCT image is the vertical or horizontal red line-layer is indicated by the yellow dot (although the yellow dot is sometimes in a hidden mode). In (**a**), graft (white arrows)-related rejection with cell infiltration (yellow arrows) was suspected. In (**b**), grafts (white arrows) were clearly shown and became sheet-like without symptoms of immune rejection at least for 6 months. White arrows: RPE grafts; dashed line circle in FAF: transplanted area.

resulted in better graft survival in vivo, which was underlain by low immunogenicity of the MHC-II deficient cells as shown in vitro. These results suggest substantial involvement of MHC-II molecules in immune response against the graft. Although in vivo rejection of *CIITA*$^{+/+}$ wild-type iPSC-RPE shown in this study was relatively mild, we observed inflammatory cells invading the retina and the choroid after transplantation. Previously, we reported more severe rejection in transplantation of wild-type miPSC-RPE (especially heterozygote miPSC-RPE), which showed high response of all lymphocytes, including CD4$^+$ T cells, by the LGIR test[27]. In vivo analysis showed *CIITA*$^{+/+}$ iPSC-RPE transplanted eyes had marked immune cell infiltration and activation around the grafted cells, while eyes transplanted with *CIITA*$^{-/-}$ miPSC-RPE or MHC-II deficient mESC-RPE had almost no infiltration and activation of the immune cells, indicating low immunogenicity of RPE cells if MHC-II is not expressed. Importantly, CD4$^+$ T cells collected from a monkey that showed severe rejection significantly responded to wild-type RPE, but not to MHC-II knockout RPE,

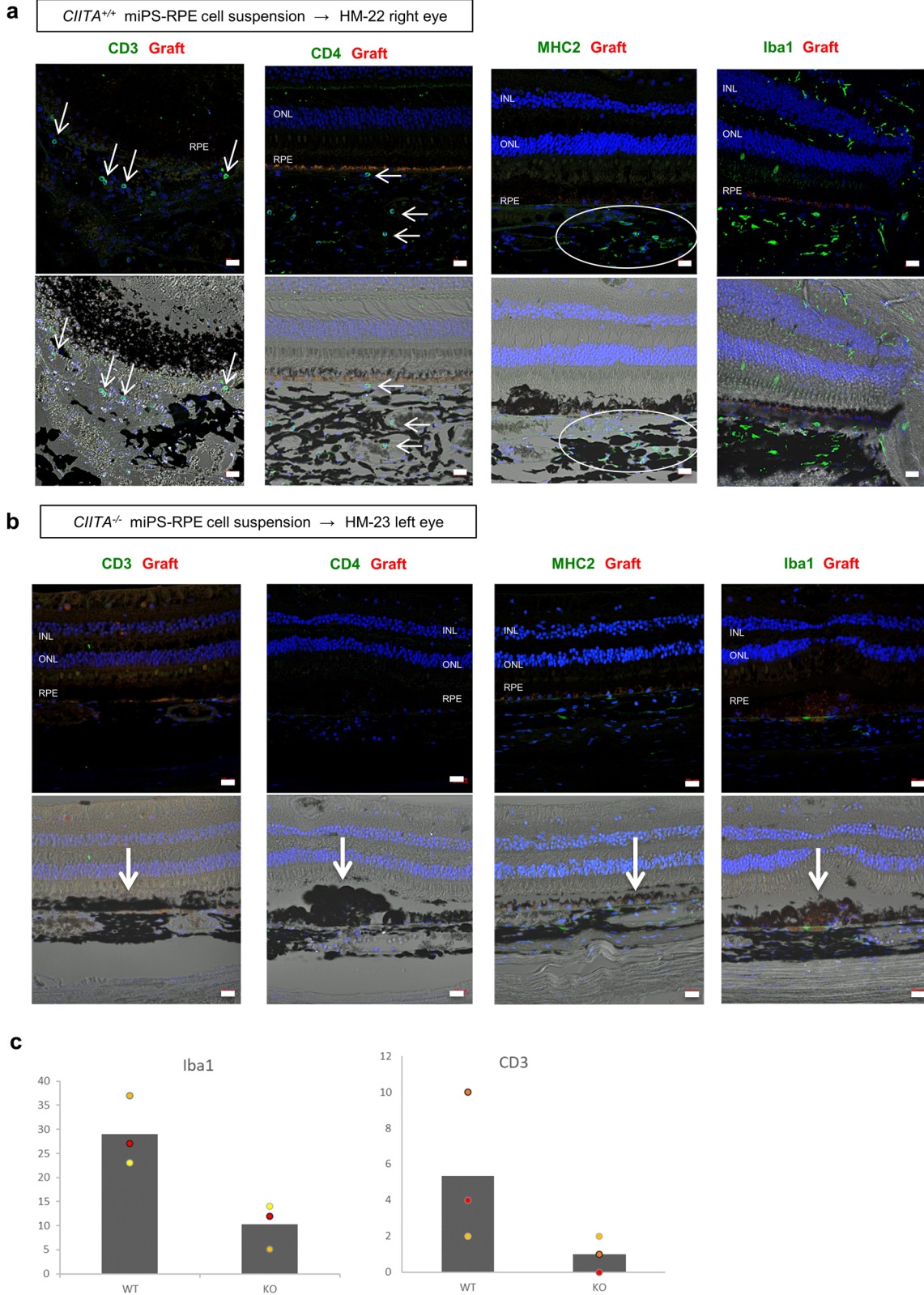

in vitro. These results suggest CD4$^+$ memory helper T cells are involved in graft rejection via the MHC-II molecules expressed on the grafts.

As mentioned earlier, transplantation of wild-type control in this study did not result in severe rejection, at a level that strongly contrasts with MHC-II-knockout transplantation. This kind of mild rejection against wild-type RPE transplantation occasionally happens as summarized in our previous

study[32]. The variability of rejection level could be due to the condition of the subjects, their immunological histories, or the surgical procedure. Interestingly, even the left and right eyes of the same monkey receiving the same surgery at different time points showed different levels of rejection[32]. To overcome these experimental/individual differences with limited number of animal subjects, in this study we provided quantification of immune cell

**Fig. 5 | IHC of inflammatory cells after transplantation of *CIITA*$^{+/+}$ or *CIITA*$^{-/-}$ iPSC-RPE cells. a** IHC staining and bright field images of HM-22 monkey transplanted with *CIITA*$^{+/+}$ miPSC-RPE allografts showed inflammatory cells such as CD3$^+$/CD4$^+$ T cells (arrows, stained in green in the left two panels), MHC-II$^+$ APCs (inside the white oval, stained in green in the third left panel) and Iba1$^+$ microglia/macrophages (stained in green in the rightmost panel) in the retina and the choroid. **b** HM-23 monkey transplanted with *CIITA*$^{-/-}$ miPSC-RPE allografts showed less CD3$^+$/CD4$^+$ T cells (stained in green in the left two panels), MHC-II$^+$ APCs (stained in green in the third left panel) and Iba1$^+$ microglia/macrophages (stained in green in the rightmost panel), with many pigmented cells (thick arrows), partially sheet-

like, in the subretinal space. PKH (red) pre-labeling of the grafts was faint after 6 months of transplantation. DAPI (blue) was used for counter staining. INL inner nuclear layer, ONL outer nuclear layer, RPE retinal pigment epithelium. Scale bars: 20 μm. **c** Quantification of invading CD3$^+$/CD4$^+$ T cells and MHC2$^+$/Iba1$^+$ microglia/macrophages after transplantation of *CIITA*$^{+/+}$ miPSC-RPE allografts (WT; monkeys HM-22, K177, S2-4) and of *CIITA*$^{-/-}$ miPSC-RPE allografts (KO; monkeys HM-23, HM-24, HM-25). IHC images used for quantification are in Supplementary Fig. 11 (*n* = 3 monkeys each). Y-axis: the frequency of iba1- or CD3-positive cells within one retinal section. Bar represents mean; *P* = 0.050 (iba1) and 0.083 (CD3) (Mann–Whitney *U*-test).

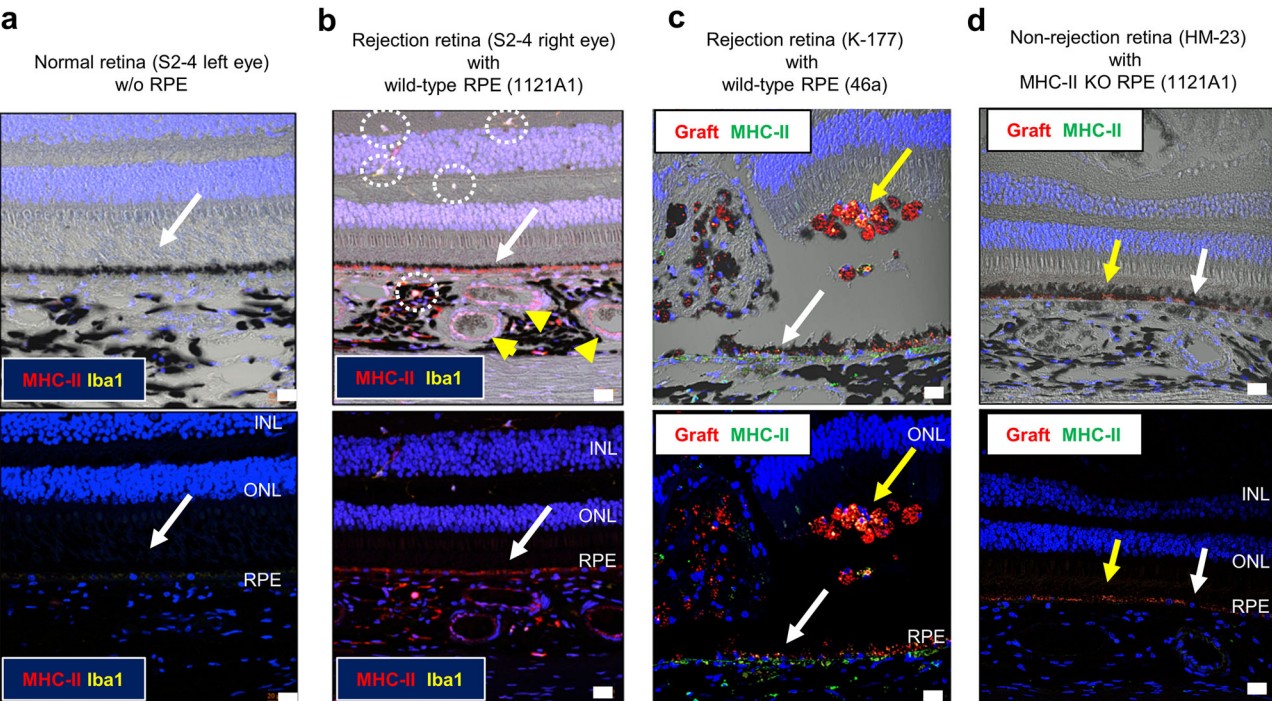

**Fig. 6 | IHC of MHC-II in host and graft RPE cells after transplantation. a** Retinal section of a normal retina (the left eye of S2-4 monkey without RPE transplantation) was stained with MHC-II (red) and iba1 (green). **b** Retinal section of the right eye of S2-4 monkey that showed rejection after transplantation with wild-type RPE (1121A1 iPSC-derived) was stained after 6 months of transplantation with MHC-II (red) and iba1 (green). **c** Retinal section of K-177 monkey that showed rejection after transplantation of PKH (red)-prelabeled wild-type RPE (46a iPSC-derived) was stained after 6 months of transplantation with MHC-II (green). **d** Retinal section of HM-23 monkey that did not show rejection after transplantation of PKH (red)-

prelabeled MHC-II knockout RPE (1121A1 iPSC-derived) was stained after 6 months of transplantation with MHC-II (green). DAPI (blue) was used for counterstaining. PKH (red) pre-labeling of the grafts was faint after 6 months of transplantation (**c**, **d**). MHC-II was detected in wild-type-transplanted (**b**, **c**) but not in normal (**a**) or MHC-II knockout-transplanted (**d**) retinas. Upper panels are merged images with bright filed that show pigmented RPE layer. White arrows: host RPE layer, yellow arrows: graft RPE, dashed line circles: double-stained cells, yellow arrowheads: blood endothelial cells. INL inner nuclear layer, ONL outer nuclear layer, RPE retinal pigment epithelium. Scale bars: 20 μm.

response (Fig. 5c), examples of rejection sites shown after wild-type transplantation that was different from MHC-II-knockout transplantation (Fig. 6), IHC of the retina that showed typical rejection after wild-type transplantation in our previous study (Supplementary Fig. 3), and an in vitro immunological assay of immune cells that once showed strong rejection against wild-type grafts becoming unresponsive to MHC-II-knockout grafts (Fig. 7c).

Although there have been no reports of gene-edited miPSC/ESC-RPE allogenic transplantation in monkeys, the potential of MHC-edited stem cells and its derivatives were reported by transplantation of human iPSC-derived endothelial cells, smooth muscle cells, or cardiomyocytes into humanized mice[16] or by xenogeneic transplantation of human ESC-RPE into rabbits[18]. Interestingly, primary uveal melanocytes and ocular melanoma cells are resistant to IFN-γ-stimulation for the expression of MHC-II and evade immune surveillance, which could be caused by so-called silencing of CIITA expression[37]. We are still investigating the cause of MHC-II

non-expression found in the mESC-RPE used in this study, but considering the results of the experiments with *CIITA*$^{-/-}$ miPSC-RPE, it should be reasonable to conclude that the low immunogenicity was due to MHC-II deficiency.

Acute allogeneic graft rejection is a major cause of early graft loss in solid organ transplantations[38]. It is thought to be mediated by host CD8$^+$ cytotoxic T lymphocytes that target graft MHC-I, for which the activity of CD4$^+$ T cells is essential[39]. As revealed in this study, MHC-II is not constitutively expressed in RPE cells, but its expression is increased under the presence of IFN-γ, which leads to rejection. Vascular endothelial cells (VECs) also express MHC-II in a similar manner[40,41] and VECs are the target cells of acute rejection, which is one of the causes of graft failure in organ transplantation[42]. In VECs, MHC-II knockout has been reported to suppress their rejection[43], which is consistent with our hypothesis.

Considering the rejection by KIR ligand mismatch in VECs[44] or the fact that NK cells were highly activated in mycoplasma-infected iPSC-RPE[45], the

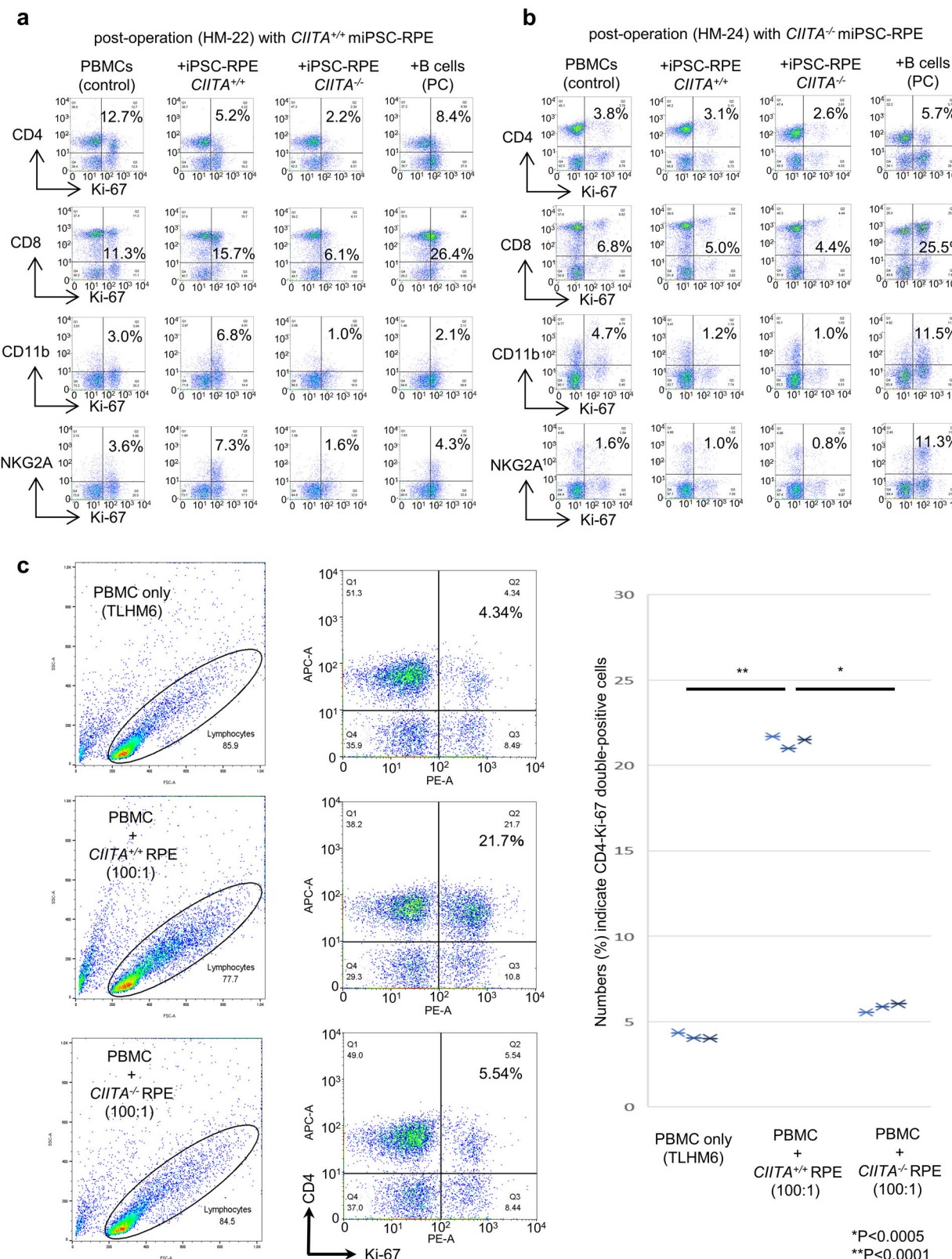

significance of MHC-I that suppresses NK cell activity must be considered. However, in the present study, NK cell infiltration was not observed in any of the monkeys, and NK cell activation was rarely observed in the rejected eyes of mismatched MHC transplants in our previous RPE transplantation study[26,27], probably reflecting the immune privileged environment of the eye, in which NK cells are suppressed at least in part by the RPE[46]. Despite the

immune privileged environment with suppressed T cell activity[21,34], slight upregulation of T cell was observed by mismatched MHC-I transplantation, which was suppressed by deletion of MHC-II (Fig. 5c). Mismatched MHC-I activates CD8[+] killer T-cells whose activity is enhanced by CD4[+] helper T-cells that is activated by MHC-II. Probably, after transient upregulation of CD8[+] killer T-cells by mismatched MHC-I, they could not get fully active

**Fig. 7 | In vitro immunological blood tests after *CIITA*⁺/⁺ or *CIITA*⁻/⁻ miPSC-RPE cell transplantation. a** The LGIR test of HM-22 monkey who received wild-type *CIITA*⁺/⁺ transplantation and showed slight rejection was performed by co-culture of its PBMC (collected 12 weeks after transplantation) with *CIITA*⁺/⁺ or *CIITA*⁻/⁻ miPSC-RPE. Allogenic monkey B95-8 B cells were used instead of RPE cells as a positive control (PC). The LGIR test with *CIITA*⁺/⁺ miPSC-RPE but not with *CIITA*⁻/⁻ miPSC-RPE showed increase in CD8⁺/Ki-67⁺ (proliferative cytotoxic T cells), CD11b⁺/Ki-67⁺ (proliferative monocytes), and NKG2A⁺/Ki-67⁺ (proliferative NK cells) compared to PBMC only (no RPE). **b** The LGIR test of HM-24 monkey who received *CIITA*⁻/⁻ (MHC-II knockout) transplantation was performed as in (**a**) with PBMC collected 8 weeks after transplantation. By either *CIITA*⁺/⁺ or *CIITA*⁻/⁻ miPSC-RPE, the proliferation of immune cells (CD4⁺, CD8⁺, CD11b⁺, and NKG2A⁺ cells) did not increase. Numbers (%) indicate double-positive cells. **c** The LGIR test of TLHM6 who had shown severe immune rejection after RPE cell transplantation in our previous study[31] was performed as in (**a**). The LGIR test with *CIITA*⁺/⁺ miPSC-RPE but not with *CIITA*⁻/⁻ miPSC-RPE showed a significant increase in proliferative CD4⁺ T cells. The plots on the left (X-axis: FSC, Y-axis: SSC) was used to define the gate according to our preliminary study to include the population of resting and proliferating lymphocytes. Statistical analyses of three independent experiments are shown on the right panel. *$P < 0.0005$, **$P < 0.0001$. $N = 3$ independent experiments; degree of freedom = 2; paired t-test; two-sided; PBMC only vs PBMC + WT RPE: $P = 8.52635E-05$ ($t$-value = $-108.290584$); PBMC + WT RPE vs PBMC + CIITA-KO RPE: $P = 0.000371324$ ($t$-value = $51.88027148$).

without CD4⁺ helper T-cells in MHC-II-knockouts. Therefore, it is possible that it may not be necessary to delete MHC-I in RPE transplantation that targets immune privileged eyes. Besides, it has been reported that instead of completely knocking out the MHC class I and II molecules, it is advantageous to leave specific MHC-I molecules to suppress NK cell activity[13,14]. There are still many unanswered questions about immune tolerance, and some reports suggest that direct recognition of CD8⁺ T cells via MHC-I is involved in immune tolerance[47], so further investigation is needed to consider the risks and benefits of MHC-I knockout strategy for transplantation.

MHC-II deficient graft for RPE transplantation has two major advantages. First, as long as immune privileged organs such as the eye are considered, in which mismatched MHC-I can be less concerned as shown in this study if MHC-II is deleted, theoretically one line is enough to treat all retinal patients, which avoids the problem of high cost and tough labor that generally accompanies transplantations of autologous, MHC matched, or gene-edited stem cell derivatives. In fact, it was hard to recruit patients for the clinical trial of MHC matched iPSC-RPE transplantation[12] because only 17% of patients with age-related macular degeneration possessed the HLA haplotype identity that fulfilled the requirement of MHC six locus concordance[48]. Second, immune rejection of MHC-II deficient RPE grafts can be controlled by local steroid administration (sub-tenon's triamcinolone acetonide injection) without systemic immunosuppressive drugs. Patients with ocular retinal disease are often elderly, which limits the use of immunosuppressants. Although the use of immunosuppressive agents should be avoided whenever possible, it is necessary for ESC-RPE transplantation while several adverse events associated with immunosuppression have been reported[5]. Adverse events with immunosuppressive drugs may be more than reported, including minor ones. If MHC-II KO RPE grafts are used for transplantation, topical steroids should be sufficient to avoid rejection and postoperative inflammation, so theoretically it would be applicable to more patients with retinal degeneration. We are currently in the process of generating RPE cells from human HLA-II knocked out iPSCs for clinical use, and are planning human clinical trials with this genome-edited line.

## Data availability
The numerical results underlying Figs. 5c and 7c are available in Supplementary Data 2. Additional data obtained during the study is available from the corresponding author upon reasonable request.

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

## Acknowledgements

We thank the technical assistance of N. Hayashi, K. Iseki, S. Fujino, K. Kawai and M. Kawahara (Laboratory for Retinal Regeneration, RIKEN Center for Developmental Biology, Kobe).

## Author contributions

M.I. and S.S. have made substantial contributions to the conception or design of the work. M.I., T.M., M.T., and S.S. were responsible for obtaining funding. M.I. and H.K. performed animal surgery. M.I., T.M., and N.S. performed an establishment of *CIITA* knockout iPSCs and iPSC-RPE. T.S. performed genotyping of the monkeys. M.I., T.M., and S.S. contributed to the main experiments, analysis and interpretation of data. M.I. and S.S. have drafted the manuscript. M.I., Y.N.F., and S.S. were responsible for critical revision of the manuscript for important intellectual content. All authors reviewed and approved the final manuscript.

## Competing interests

The authors declare the following competing interests: M.T. received research funding from AMED. S.S. and Y.N.F. received research funding from JSPS KAKENHI Grant-in-Aid for Scientific. M.T. and S.S. are employees of Vision Care Inc. T.M., N.S., and Y.N.F. are employees of VC Cell Therapy Inc. The remaining authors declare no conflict of interest. All authors attest that they meet the current ICMJE criteria for authorship.
