## [Peer Review File · Communications Medicine]

Reviewers' comments:

Reviewer #1 (Remarks to the Author):

The authors present a very interesting study with preclinical relevance for the retinal and stem cell therapy community. Although currently there are multiple studies aiming to engineer stem cells to avoid immune response, the study presented by Ishida et al with edited retinal pigment epithelium (RPE) derived from monkey-induced pluripotent stem cells lacking MHC-II (miPSC-RPE CIITA^{-/-}) show valuable survival and infiltration data in the monkey eye for the advancement of PSC-RPE as a therapeutic strategy with minimal use of immunosuppression. Results and figures are well presented and described.

I have the following comments for the authors that are important to be addressed:

Major Comments:

1. Immunofluorescence images and flow plots are shown in poor resolution. Please increase the resolution of them to make contents clearer and readable.
2. Since immune rejection/infiltration is a critical measure in the whole study, quantification/measurement of the infiltrates pointed at the OCT images is needed throughout the manuscript.
3. Also, please include an example of a not rejected (without graft) and of fully rejected graft (fundus, OCT, quantification) in supplementary figure for reference.
4. Could you please explain why in Figure 1 mESC-RPE are engrafted as cell sheet and then cell injection is changed to suspension?
5. Figure 2e: it is important to have CIITA^{+/+} as positive control in the flow plots, so please include them.
6. From the pictures shown in Figure 3 it is not clear that CIITA^{+/+} miPSC-RPE show a decreased immune reaction compared to CIITA^{-/-} miPSC-RPE; also all grafts are small and seem to be equally surviving in both cases unless demonstrated by other assays (e.g. apoptosis staining), so conclusions stated in lines 165-168 need to be tuned down. Please also re-phrase lines 269-270 of the discussion.
7. Figure 4a: for comparison please add H&E staining of the graft region for CIITA^{+/+} miPSC-RPE injected cells.

8. In general grafted cells (red: PKH-positive) are not seen clearly in the immunofluorescence stainings in Figures 4, S5 and S6 (include it in S6c), so please increase contrast or re-stain with an HLA-allele specific marker so they can be clearly distinguished.

9. Related to that, could you please explain how “a lot of expanded pigmented RPE cells” (line 182) is measured and compared to CIITA^{+/+} grafts in the images presented? As it is shown this cannot be stated.

10. In order to conclude that there are no inflammatory signs/cells in injected CIITA^{-/-} vs CIITA^{+/+} miPSC-RPE, please include a bigger area of image similar scale as the H&E presented in Figure 4b with CD3, CD4, MHC2 and Iba1 markers for both CIITA^{-/-} and CIITA^{+/+} injected cells.

11. Figure 6a and b: what is PBMC (control)? If it is PBMC-only, could you explain the elevated amounts of CD4 and CD8 compared to iPSC-RPE CIITA^{+/+}?

12. It would be very interesting to evaluate the presence of B cells and specific antibodies against the graft at 6 months in both CIITA^{-/-} and CIITA^{+/+} injected animals. This could be addressed firstly by staining sections with CD19 or CD20, and second, by incubating serums with wild-type miPSC-RPE and assessing anti-monkey antibodies by flow.

13. In the discussion, could you please elaborate more on i) why NK cell infiltration/activation is not observed in the monkeys analysed with mismatched MHC grafts without immunosuppression (lines 308-310)? and ii) why the CIITA^{-/-} miPSC-RPE cells survive with no detected immune infiltration despite mismatched MHC-I (lines 313-315)?

14. Could you clarify why you state that “one line is enough to treat all retinal patients” with MHC-II deficient cells (line 324)? Is it because assuming that no immune rejection would be elicited at all in a human setting?

Minor Comments:

1. Typo in line 123: correct to “rejection”.

2. Typo in line 232: correct to “signs”.

Reviewer #2 (Remarks to the Author):

The purpose of this paper was to investigate whether suppression or deletion of MHC-II molecules can decrease the immunogenicity of iPSC-derived RPE allografts. The study was performed in cynomolgus monkeys. This was based on an initial discovery that iPSC-RPE allograft transplants

were not rejected in one monkey recipient that did not express MHC-II molecules. Therefore, 2 monkey iPSC lines (miPSCs) with a knockout of the gene coding class II MHC transactivator (CIITA) were created by Crispr-Cas9 gene editing. These CIITA^{-/-} lines (and control iPSC CIITA^{+/+} lines) were differentiated into RPE. CIITA^{-/-} line-derived RPE failed to express MHC-II even after stimulation with IFN- γ . iPSC-derived RPE cells were transplanted as allografts into 4 monkeys. (no MHC-II match). Three monkeys received CIITA^{-/-} RPE (6 eyes), one monkey received wild-type CIITA^{+/+} RPE (2 eyes). One of these monkeys received CIITA^{-/-} RPE treated with ROCK inhibitor to promote engraftment. The monkey receiving the wildtype RPE showed some RPE-graft-related rejection as observable by OCT and histology, whereas . CIITA^{-/-} RPE could survive for at least 6 months without any signs of rejection. Sections through grafts were analyzed with markers for immune cells and MHC-II expression. An in vitro blood test (LGIR) comparing lymphocyte reaction against wildtype and CIITA^{-/-} RPE cells showed that CIITA^{-/-} RPE cells did not elicit a response.

General comments:

This is a well-written paper with important findings. The results of the study could lead to better allograft survival without needing immunosuppressive drugs; and could also be important for other types of transplants. The next step will be to create MHC-knockout human iPSCs.

However, due to this being a monkey study, the N of the experiments is low.

Some minor issues: It looks like there was no clear additional effect of ROCK-inhibitor treatment of one transplant due to the low N. Another issue: no way to identify transplanted cells.

Specific comments:

Methods:

p. 16, line 349: please mention the sex of the monkeys. Where both males and females used, or only one sex, and why?

Figure 1 contains too many panels, resulting in low resolution. The figure should be split in two. Proposal: move panels g - i to a new figure.

Otherwise the figures are excellent.

Figure legends of Fig. 5:

p. 30, line 765: "Lower panels are bright filed images" – should be "Upper panels are bright field images"

Reviewer #3 (Remarks to the Author):

In reviewing the study on MHC-II knockout pluripotent stem cell-derived RPE, the approach of leveraging gene editing to modulate immune molecule expression is a meaningful stride in reducing immune rejection impacts in cell transplantation. This could be pivotal in minimizing immunosuppressant side effects and broadening induced pluripotent stem cell sources.

1. The novelty of the research is somewhat overshadowed by existing literature proposing similar immuno-compatibility enhancements in iPSCs via CRISPR-Cas9, targeting HLA genes[doi:10.1038/nbt.3860].
2. The paper primarily focuses on the immunogenicity and survival of altered RPE cells, yet it falls short in discussing the impact on visual function improvement, which is crucial for ophthalmologic applications. Detailed assessments of visual function post-transplantation would greatly enhance the study's relevance and application in clinical settings.
3. The functional implications of MHC-II gene knockout warrant further exploration, particularly considering the intricate cellular interactions necessary for vision. MHC-II's role in cell communication might suggest potential functional changes or losses post-knockout.
4. There appears to be a discrepancy between the findings of this study and previous research [DOI:<https://doi.org/10.1016/j.stemcr.2021.02.021>, DOI:<https://doi.org/10.1016/j.stemcr.2022.09.014>] indicating no significant rejection signs in MHC-mismatched transplantations without immune suppression. Clarifying methodological differences or interpretations could resolve this inconsistency and strengthen the study's conclusions.

In summary, while the research presents a promising strategy in transplantation, it would benefit from a more comprehensive examination of functional outcomes and a clearer differentiation from existing methodologies.

Point-to-point response:

We thank the reviewers for their careful comments. The following are our point-to-point response indicated in blue. The corresponding sentences in the main text are indicated in red. We hope this version of our manuscript satisfy the reviewers' concerns.

Reviewers' comments:

Reviewer #1 (Remarks to the Author):

The authors present a very interesting study with preclinical relevance for the retinal and stem cell therapy community. Although currently there are multiple studies aiming to engineer stem cells to avoid immune response, the study presented by Ishida et al with edited retinal pigment epithelium (RPE) derived from monkey-induced pluripotent stem cells lacking MHC-II (miPSC-RPE CIITA-/-) show valuable survival and infiltration data in the monkey eye for the advancement of PSC-RPE as a therapeutic strategy with minimal use of immunosuppression. Results and figures are well presented and described.

I have the following comments for the authors that are important to be addressed:

Major Comments:

1. Immunofluorescence images and flow plots are shown in poor resolution. Please increase the resolution of them to make contents clearer and readable.

Thank you for the comment. We agree some of the figures were over crowded, which made the resolution of each image low. For better resolution of the immunostaining images, we divided original Fig1 into two (new Fig1 and Fig2), and also moved original Fig4b to supplementary new FigS9b.

2. Since immune rejection/infiltration is a critical measure in the whole study, quantification/measurement of the infiltrates pointed at the OCT images is needed throughout the manuscript.

We appreciate the comment. In the revision, we added quantification of infiltrating iba1-positive microglia/macrophages and CD3-positive T-cells in Fig5c, and disclosed all the images used for the quantification in Fig S10. We described this in main text page 9 line 194-196 (indicated in red).

3. Also, please include an example of a not rejected (without graft) and of fully rejected graft (fundus, OCT, quantification) in supplementary figure for reference.

Thank you, we agree this data should be informative. We added immunostaining images of inflammatory cells in normal retina without graft, and in fully rejected retina, in supplementary new Fig. S2 (cited at main text page 6, indicated in red). The OCT images of fully rejected retina have already been shown in our previous study (ref 33). We referred to this in the main text page 5 line 100-101 (indicated in red).

4. Could you please explain why in Figure 1 mESC-RPE are engrafted as cell sheet and then cell injection is changed to suspension?

Thank you for the comment. The reason for using cell sheet first and then moved to cell suspension in this study was according to our clinical strategy. Although our first clinical trial for iPSC-RPE transplantation was realized in a form of RPE-sheet transplantation (ref 3), subretinal sheet transplantation was generally invasive, so afterwards we moved to a less invasive form such as cell suspension injection. We described this in the revision at the end of page 5-6 line 107-111 (indicated in red).

5. Figure 2e: it is important to have CIITA^{+/+} as positive control in the flow plots, so please include them.

Thank you for the comment. We added flow plots in new supplementary Fig. S6 showing CIITA^{+/+} RPE cells express MHC-II in response to IFN γ -stimulation, and described in main text page 7 line 148 (indicated in red).

6. From the pictures shown in Figure 3 it is not clear that CIITA^{+/+} miPSC-RPE show a decreased immune reaction compared to CIITA^{-/-} miPSC-RPE; also all grafts are small and seem to be equally surviving in both cases unless demonstrated by other assays (e.g. apoptosis staining), so conclusions stated in lines 165-168 need to be tuned down. Please also re-phrase lines 269-270 of the discussion.

Thank you for the comment. We agree that for some reason the wildtype CIITA^{+/+} transplantation in this study did not result in extreme immune reaction as expected (for example in our previous study, allograft transplantation generally gave more obvious immune reaction, as we added in new supplementary Fig. S2). Thus, we tuned down our conclusion as follows (indicated in red in the main text page 8 line 175 and page 13 line 277):

page 8 line 175

(original) transplantation with CIITA^{-/-} iPSC-RPE cells allows graft survival

(re-phrased to) transplantation with CIITA^{-/-} iPSC-RPE cells could be efficient for graft

survival

page 13 line 277

(original) promoting graft survival

(re-phrased to) gave better graft survival

7. Figure 4a: for comparison please add H&E staining of the graft region for CIITA^{+/+} miPSC-RPE injected cells.

Thank you for the comment. We added H&E staining of the graft region for CIITA^{+/+} miPSC-RPE transplantation in new supplementary Fig S9a.

8. In general grafted cells (red: PKH-positive) are not seen clearly in the immunofluorescence stainings in Figures 4, S5 and S6 (include it in S6c), so please increase contrast or re-stain with an HLA-allele specific marker so they can be clearly distinguished.

Thank you for the comment. After going through the IHC images, including many transplanted eye sections from our previous study, we admit PKH pre-labeling, although used to trace the graft, generally becomes faint after 6 months of transplantation, thus was not appropriate to point the grafts by PKH pre-labeling in this study which had all the IHCs performed after 6 months of transplantation (manufacturer's warranty for PKH-tracking was for 120 days, and we usually see bright red grafts if sacrificed within 3 month of transplantation). We added this explanation in the legends of new Figs 2d, 5 and 6 (indicated in red) instead of increasing the contrast.

9. Related to that, could you please explain how "a lot of expanded pigmented RPE cells" (line 182) is measured and compared to CIITA^{+/+} grafts in the images presented? As it is shown this cannot be stated.

We appreciate the comment. Without PKH pre-labeling as mentioned above, although it still gave faint clues, the way we identified grafts was by their characteristic appearance: generally darker than host RPEs, tended to form not a monolayer but thick multilayer, sometimes overlaying the host RPEs and sometimes migrating beneath the host RPE, eventually integrate and become sheet-like (added in page 6 line 117-121, indicated in red). While we can tell from our experience that this qualification should be true, we admit the subjectiveness cannot be denied. Besides, as the images presented in original Fig 4c (new Fig 5b) were not analyzed by proliferation markers, and also as our interpretation of engraftment was RPE-transplants becoming sheet-like after integration (Fig 2a legend), we agree that we cannot tell from these images whether they represented expansion and/or integration of graft RPEs. Thus, we would like to delete the sentence pointed out by the reviewer (line 182 in the original manuscript), and focus on the fact

that MHC-II-knockout potentially suppresses immune activity against allografts. The surrounding sentences after deletion are indicated in red in page 9 line 187-191. This modification should make the conclusion of this study more clear: MHCII-KO transplants give less emergence of inflammatory cells (which is now quantified in the revised Fig 5c and supplemental Fig S10, main text page 9 line 194-196 indicated in red).

10. In order to conclude that there are no inflammatory signs/cells in injected CIITA^{-/-} vs CIITA^{+/+} miPSC-RPE, please include a bigger area of image similar scale as the H&E presented in Figure 4b with CD3, CD4, MHC2 and Iba1 markers for both CIITA^{-/-} and CIITA^{+/+} injected cells.

Thank you for the comment. We do have these IHC images with lower magnification that covers wider area, however the resolution is too low to detect positive cells. Instead, we have collected IHC images from 6 monkeys (3 of WT-transplanted and 3 of MHCII-KO-transplanted) with higher resolution for quantification (new Fig 5c), and disclosed all the images in supplemental new Fig S10 (main text page 9 line 194-196, indicated in red). We hope this quantification is sufficient for the comparison between WT- and MHC-KO-transplanted, and would lead to the conclusion.

11. Figure 6a and b: what is PBMC (control)? If it is PBMC-only, could you explain the elevated amounts of CD4 and CD8 compared to iPSC-RPE CIITA^{+/+}?

Thank you for the comment. "PBMC control" meant culture of PBMC without co-culture with graft RPE cells. We clarified this by replacing "PBMC control" with "PBMC only" in Fig7a and Fig7b. The reason for "PBMC only" having higher immune activity compared to the co-culture with RPE cells was due to the immune suppressive characteristics of the RPE cell (ref 45). We added this explanation in the main text (page 11 line 238-240, indicated in red).

12. It would be very interesting to evaluate the presence of B cells and specific antibodies against the graft at 6 months in both CIITA^{-/-} and CIITA^{+/+} injected animals. This could be addressed firstly by staining sections with CD19 or CD20, and second, by incubating serums with wild-type miPSC-RPE and assessing anti-monkey antibodies by flow.

Thank you for the comment. We added a staining of CD20 in new supplementary Fig S2, for a monkey that showed severe immune rejection (Fig 7c, ref 33). For the same monkey, serum incubation with miPSC-RPE has already been performed in our previous study (ref 33), which detected specific antibodies against graft RPE.

13. In the discussion, could you please elaborate more on i) why NK cell infiltration/activation is not observed in the monkeys analysed with mismatched MHC grafts without

immunosuppression (lines 308-310)? and ii) why the CIITA^{-/-} miPSC-RPE cells survive with no detected immune infiltration despite mismatched MHC-I (lines 313-315)?

Thank you for the comment. We agree this part was difficult to read. i) we interpreted suppressed NK cell activity was due to the immune-privileged environment of the eye, as we previously revealed one of the mechanisms of NK-cell suppression in the eye (ref 46). We added this reference and explained in page 14 line 322-323 (indicated in red). ii) again, due to the immune-privileged environment of the eye, basically T-cells activities are suppressed (refs 19, 45) but can slightly be upregulated by mismatched MHC-I as shown by CIITA^{+/+} RPE transplantation in this study. Generally, MHC-I activates killer T-cells whose activity is enhanced by helper T-cells that is activated by MHC-II. Thus, we consider even though mismatched MHC-I has caused transient upregulation of killer T-cells, they could not get fully active without the support of helper T-cells that should have been activated by MHC-II. We added this explanation in page 14-15 line 323-330 (indicated in red).

(ref 46) Sugita S, Makabe K, Iwasaki Y, Fujii S, Takahashi M. Natural Killer Cell Inhibition by HLA-E Molecules on Induced Pluripotent Stem Cell-Derived Retinal Pigment Epithelial Cells. *Investigative Ophthalmology & Visual Science* 59, 1719-1731. doi: 10.1167/iovs.17-22703 (2018).

14. Could you clarify why you state that “one line is enough to treat all retinal patients” with MHC-II deficient cells (line 324)? Is it because assuming that no immune rejection would be elicited at all in a human setting?

Thank you for the comment. As mentioned above, at least in the immune privilege environment of the eye, the results of the present study showed MHC-I mismatch can be less concerned. Thus, as long as MHC-I mismatch is not an issue unlike other organs, we meant MHC-II-KO might be sufficient to overcome the difference in MHC-I between the recipient and the graft. We clarified this in page 15 line 331 and line 340-342 (indicated in red).

Minor Comments:

1. Typo in line 123: correct to “rejection”. **Corrected.**

2. Typo in line 232: correct to “signs”. **Corrected.**

Reviewer #2 (Remarks to the Author):

The purpose of this paper was to investigate whether suppression or deletion of MHC-II molecules can decrease the immunogenicity of iPSC-derived RPE allografts. The study was performed in cynomolgus monkeys. This was based on an initial discovery that iPSC-RPE allograft transplants were not rejected in one monkey recipient that did not express MHC-II molecules. Therefore, 2 monkey iPSC lines (miPSCs) with a knockout of the gene coding class II MHC transactivator (CIITA) were created by Crispr-Cas9 gene editing. These CIITA^{-/-} lines (and control iPSC CIITA^{+/+} lines) were differentiated into RPE. CIITA^{-/-} line-derived RPE failed to express MHC-II even after stimulation with IFN- γ . iPSC-derived RPE cells were transplanted as allografts into 4 monkeys. (no MHC-II match). Three monkeys received CIITA^{-/-} RPE (6 eyes), one monkey received wild-type CIITA^{+/+} RPE (2 eyes). One of these monkeys received CIITA^{-/-} RPE treated with ROCK inhibitor to promote engraftment. The monkey receiving the wildtype RPE showed some RPE-graft-related rejection as observable by OCT and histology, whereas . CIITA^{-/-} RPE could survive for at least 6 months without any signs of rejection. Sections through grafts were analyzed with markers for immune cells and MHC-II expression. An in vitro blood test (LGIR) comparing lymphocyte reaction against wildtype and CIITA^{-/-} RPE cells showed that CIITA^{-/-} RPE cells did not elicit a response.

General comments:

This is a well-written paper with important findings. The results of the study could lead to better allograft survival without needing immunosuppressive drugs; and could also be important for other types of transplants. The next step will be to create MHC-knockout human iPSCs.

However, due to this being a monkey study, the N of the experiments is low.

Some minor issues: It looks like there was no clear additional effect of ROCK-inhibitor treatment of one transplant due to the low N.

Thank you for the comment. As we knew ROCK-inhibitor promotes engraftment (ref 31), the intention of using ROCK-inhibitor here was to assess immune response in a condition with better engraftment, which should be a more severe condition if immune rejection against the graft matters. We clarified this in page 8 line 171-172 (indicated in red). Otherwise, we do not think ROCK-inhibitor has any immunological effects.

Another issue: no way to identify transplanted cells.

Thank you for the comment. Although PKH pre-labeling was used to trace the graft, generally it becomes faint after 6 months of transplantation, thus was not appropriate to point the grafts by PKH pre-labeling in this study which had all the IHCs performed after

6 months of transplantation (manufacturer's warranty for PKH-tracking was for 120 days, and we usually see bright red grafts if sacrificed within 3 month of transplantation). We added this explanation in the legends of new Figs 2d, 5 and 6 (indicated in red). Although PKH pre-labeling still gave faint clues, the way we identified grafts was by their characteristic appearance: generally darker than host RPEs, tended to form not a monolayer but thick multilayer, sometimes overlaying the host RPEs and sometimes migrating beneath the host RPE, eventually integrate and become sheet-like. We added this description in page 6 line 117-121 (indicated in red).

Specific comments:

Methods:

p. 16, line 349: please mention the sex of the monkeys. Where both males and females used, or only one sex, and why?

Thank you for the comment. To avoid immune rejection due to male specific histocompatibility antigen coded on the Y-chromosome (H-Y antigen), all the monkeys (both donors and recipients) used in this study were male. We added this information in the Methods section (page 17 line 373-375, indicated in red).

Figure 1 contains too many panels, resulting in low resolution. The figure should be split in two.

Proposal: move panels g - i to a new figure.

Otherwise the figures are excellent.

Thank you. We followed the suggestion and moved Fig 1f-i to new Fig 2a-d.

Figure legends of Fig. 5:

p. 30, line 765: "Lower panels are bright filed images" – should be "Upper panels are bright field images"

Corrected.

Reviewer #3 (Remarks to the Author):

In reviewing the study on MHC-II knockout pluripotent stem cell-derived RPE, the approach of leveraging gene editing to modulate immune molecule expression is a meaningful stride in reducing immune rejection impacts in cell transplantation. This could be pivotal in minimizing immunosuppressant side effects and broadening induced pluripotent stem cell sources.

1. The novelty of the research is somewhat overshadowed by existing literature proposing similar immuno-compatibility enhancements in iPSCs via CRISPR-Cas9, targeting HLA genes[doi:10.1038/nbt.3860].

The existing literature pointed by the reviewer is about transplantation of cells that do not express MHC-II, and shows the efficacy of leaving class I HLA-E while deleting the rest of MHC-I. On the other hand, our study targets MHC-II expressing transplants, and has shown rejection could be avoided at least in immune privileged organs like the eye, if MHC-II was deleted. Our strategy leaves MHC-I intact, which should be advantageous against infection and also for suppressing NK cell related rejection. We had this discussion in page 15 line 332-334.

2. The paper primarily focuses on the immunogenicity and survival of altered RPE cells, yet it falls short in discussing the impact on visual function improvement, which is crucial for ophthalmologic applications. Detailed assessments of visual function post-transplantation would greatly enhance the study's relevance and application in clinical settings.

In this study, all the transplantation were performed in normal monkeys but not in disease models, because the purpose of the study was to establish a transplantation strategy for controlling immune rejection in a specialized immune environment of the eye, which is a very important factor for successful cell transplantation. For this reason, assessments of visual function post-transplantation were not included in the design of the study.

3. The functional implications of MHC-II gene knockout warrant further exploration, particularly considering the intricate cellular interactions necessary for vision. MHC-II's role in cell communication might suggest potential functional changes or losses post-knockout.

The function of MHC-II is to present antigens and activate CD4+ helper T-cells. At least in an immune privileged organ like the eye with rather suppressed immune activity, so far we do not find disadvantageous effects for vision by knocking out MHC-II. Regarding the immune cell communication related to MHC-II deletion, we added a discussion in page 14-15 line 323-330 (indicated in red).

4. There appears to be a discrepancy between the findings of this study and previous research [DOI:<https://doi.org/10.1016/j.stemcr.2021.02.021>, DOI:<https://doi.org/10.1016/j.stemcr.2022.09.014>] indicating no significant rejection signs in MHC-mismatched transplantations without immune suppression. Clarifying methodological differences or interpretations could resolve this inconsistency and strengthen the study's conclusions.

The previous research pointed out by the reviewer is about cell transplantation of retinal

cells that have little expression of MHC-II, unlike RPE cells that express MHC-II in response to IFN-g as described in this study. We clarified this in page 3 line 56 (indicated in red) by citing the two papers pointed by the reviewer.

In summary, while the research presents a promising strategy in transplantation, it would benefit from a more comprehensive examination of functional outcomes and a clearer differentiation from existing methodologies.

Reviewers' comments:

Reviewer #1 (Remarks to the Author):

The authors did a good job addressing most of the concerns raised. Thank you for providing infiltration quantifications, they do strengthen the conclusions of the study.

However, I would like to still point out the following:

1.(Original comment 1): Immunofluorescence images and flow plots are shown in poor resolution. Please increase the resolution of them to make contents clearer and readable.

Images are still shown in very poor resolution, not sure if it is due to the pdf compression from the journal's portal, but it would be appreciated that resolution of the figure images is increased to 300ppi for final publication.

2.(Original comment 7): Figure 4a: for comparison please add H&E staining of the graft region for CIITA^{+/+} miPSC-RPE injected cells.

Thank you for adding H&E staining of the graft region for CIITA^{+/+} miPSC-RPE transplantation in new supplementary Fig S9a. However, in CIITA^{+/+} there are areas that look exactly the same as the ones pointed out in CIITA^{-/-} (shown as thick multilayers) that are not indicated with arrows. Could the authors explain why these are not indicated in the CIITA^{+/+} H&E picture? Based on these two examples, it is not clear that there is a graft survival difference between the two miPSC-RPE cell types. Additional H&E images where differences in engrafted miPSC-RPE are clearer would be appreciated.

3.(Original comment 8): In general grafted cells (red: PKH-positive) are not seen clearly in the immunofluorescence stainings in Figures 4, S5 and S6 (include it in S6c), so please increase contrast or re-stain with an HLA-allele specific marker so they can be clearly distinguished.

Thank you for the provided explanation. I appreciate that the quality of the PKH-label was not ideal after 6 months. However, given the importance of the conclusions of the manuscript based on graft survival it is critical to re-stain respective figures and especially Figures 2D, 5 and 6 with an HLA-allele specific marker or with some other specific marker, so that the grafted cells can be convincingly distinguished.

4.(Original comment 9): Please correct sentence in lines 191-192 for “Taken together, the eyes transplanted with MHC-II knockout-RPE cells showed decreased rejection by IHC evaluations unlike wild-type transplantation”.

Reviewer #2 (Remarks to the Author):

In this revised manuscript version, the authors have addressed most of the reviewers' concerns. However, the figures appear to be reduced too much in resolution, even when downloading the source files. They look fuzzy when zooming in to the size of a computer screen, especially Figure 1 (only 510x360 pixels), Figure 4 (720x738 pixels), and Figure 6 (644x360 pixels). Apparently Figure 1 was split up without going back to the original resolution files. It is a pity because otherwise they would be great figures.

Also, the order of the supplemental figures does not correspond to their citations in the text.

Specific comments:

Introduction, p. 3, 2nd paragraph: “unlike retinal cells that express little MHC class II ...” – replace “little” with “few”

Discussion, p. 11, 1st line: “gave better graft survival” – proposal: “resulted in better graft survival”

There is a typo in the figure legend of Figure S7 (f) “bright filed”

In summary, the text revisions appear to be sufficient, but the figure quality (resolution) needs to be improved.

Reviewer #3 (Remarks to the Author):

In the current study, the author introduced interesting findings of PRE transplantation in monkeys, which could be important for preclinical study. Deficiency or knockout of MHCII in RPE cells could avoid immune rejection and improve graft survival in monkeys. My main concern of the paper is that

N was relatively low, due to experimental animals, which affected reliability. Overall, the study provided valuable resources for current understanding of PRE transplantation.

The responses of reviewers' comments were in-point, and I believe that the quality of the paper has been improved.

Point-to-point response:

We thank the reviewers for their careful comments. The following are our point-to-point response indicated in blue. The corresponding sentences in the main text are highlighted. We hope this version of our manuscript satisfy the reviewers' concerns.

Reviewer #1 (Remarks to the Author):

The authors did a good job addressing most of the concerns raised. Thank you for providing infiltration quantifications, they do strengthen the conclusions of the study.

However, I would like to still point out the following:

1.(Original comment 1): Immunofluorescence images and flow plots are shown in poor resolution. Please increase the resolution of them to make contents clearer and readable.

Images are still shown in very poor resolution, not sure if it is due to the pdf compression from the journal's portal, but it would be appreciated that resolution of the figure images is increased to 300ppi for final publication.

Response: Please excuse us for keep submitting low resolution images. It seemed like we had a trouble in TIF conversion, which is now solved. We aimed 600dpi. We hope the resolution is now sufficient.

2.(Original comment 7): Figure 4a: for comparison please add H&E staining of the graft region for CIITA+/+ miPSC-RPE injected cells.

Thank you for adding H&E staining of the graft region for CIITA^{+/+} miPSC-RPE transplantation in new supplementary Fig S9a. However, in CIITA^{+/+} there are areas that look exactly the same as the ones pointed out in CIITA^{-/-} (shown as thick multilayers) that are not indicated with arrows. Could the authors explain why these are not indicated in the CIITA^{+/+} H&E picture? Based on these two examples, it is not clear that there is a graft survival difference between the two miPSC-RPE cell types. Additional H&E images where differences in engrafted miPSC-RPE are clearer would be appreciated.

Response: Thank you for the comment. We agree it was confusing not pointing engrafted area for CIITA^{+/+} transplantation. In the revised Supplemental information, we indicated the grafts by arrows in Fig S9a as in Fig S9b. One of the limitations of this study was that transplantation of WT-control did not result in severe rejection, at a level that strongly contrasts with MHCII-KO-transplantation. This kind of not severe but mild rejection against WT-RPE transplantation occasionally happens as summarized in our previous study (ref 32):

Int. J. Mol. Sci. 2020, 21(9), 3077; <https://doi.org/10.3390/ijms21093077> (Table S1)

Supplemental Table 1.**Supplemental Table 1. Summary of the *in vivo* RPE cell transplantation in monkeys**

Name	Operated eye	Graft RPE cells (human RPE)	Drugs	IHC evaluation	Rejection*	Complications
HM-1	R	Cell suspension	IVTA	6 months	+	None
	L	Cell suspension	IVTA	5 months 1 week	++	None
HM-2	R	Cell suspension	IVTA	3 months 1 week	+	None
	L	Cell suspension	IVTA	3 months	+++	None
HM-3	R	Cell suspension	IVTA	3 months	+	None
HM-4	R	Cell suspension	None	6 months	+	ERM
HM-5	R	Cell suspension	IVTA Systemic CsA	3 months	-	None
HM-6	R	Cell suspension	IVTA + STTA	8 months 1 week	±	None
	L	Cell suspension	IVTA + STTA	4 months 1 week	±	None
HM-7	R	Cell suspension	IVTA + STTA	3 months	±	Macular damage**
HM-8	R	Cell suspension	IVTA	5 months	+	ERM
	L	Cell suspension	IVTA	4 months 2 weeks	+	ERM, endophthalmitis

*Grading of the RPE graft-related immune rejections was as follows: non (-), slight (±), mild (+), moderate (++), and severe (+++). **Macular damage due to operation error (reference 21 in main text). CsA – Cyclosporine A. IVTA; intravitreal triamcinolone acetonide, STTA; sub-Tenon triamcinolone acetonide, ERM; epiretinal membrane.

We speculate the variability of rejection level was due to the condition of the subjects, their immunological histories, the surgical procedure, and so on. Interestingly, even the left and right eyes of the same monkey receiving the same surgery at different time points showed different levels of rejection. To overcome these experimental/individual differences with limited number of animal subjects, we provided quantification of immune cell response (Fig 5c: added for the first revision in response to reviewer's comment), examples of rejection sites shown after WT-transplantation where differences from MHCII-KO-transplantation were clearer (Fig 6), IHC of the retina that showed typical rejection after WT-transplantation in our previous study (Fig S3: added for the first revision in response to reviewer's comment), and an *in vitro* immunological assay of immune cells that once showed strong rejection against WT-grafts becoming unresponsive to MHCII-KO-grafts (Fig 7c).

We added this statement in the main text (Limitation of the study, page 19, highlighted).

3.(Original comment 8): In general grafted cells (red: PKH-positive) are not seen clearly in the immunofluorescence stainings in Figures 4, S5 and S6 (include it in S6c), so please increase contrast or re-stain with an HLA-allele specific marker so they can be clearly distinguished.

Thank you for the provided explanation. I appreciate that the quality of the PKH-label was not ideal after 6 months. However, given the importance of the conclusions of the manuscript based on graft survival it is critical to re-stain respective figures and especially Figures 2D, 5 and 6 with an HLA-allele specific marker or with some other specific marker, so that the grafted cells can be convincingly distinguished.

Response: We appreciate this comment but to the best to our knowledge, monkey MHC allele specific antibody that distinguishes host and graft MHCs (Table S1), or any other specific marker that distinguishes host and graft RPE, may not be available. So far, there's no antibody raised for immunostaining of Cynomolgus monkey MHCs, and most of the researches on Cynomolgus monkey models use Human MHC antibodies as substitutes. Prior to this study, we evaluated several Human MHCII antibodies and found one that sufficiently recognized Cynomolgus monkey MHCII, which was used in this study. If it was xenograft transplantation with Human-RPE transplanted into Cynomolgus monkeys, there was a way to distinguish host and graft, but in the case of allograft transplantation as in this study, there was no way besides PKH-labeling. We admit the criteria of engraftment in this study is ambiguous, but we would like to complement the ambiguousness by providing quantification of immune cell response (Fig 5c: added for the first revision in response to reviewer's comment), examples of rejection sites shown after WT-transplantation that was different from MHCII-KO-transplantation (Fig 6), IHC of the retina that showed typical rejection after WT-transplantation in our previous study (Fig S3: added for the first revision in response to reviewer's comment), and an *in vitro* immunological assay of immune cells that once showed strong rejection against WT-grafts becoming unresponsive to MHCII-KO-grafts (Fig 7c). We hope these lines of

evidence would lead to the conclusion of the manuscript.

4.(Original comment 9): Please correct sentence in lines 191-192 for "Taken together, the eyes transplanted with MHC-II knockout-RPE cells showed decreased rejection by IHC evaluations unlike wild-type transplantation".

Response: Thank you for pointing out. We corrected the sentence as follows:

"Taken together, IHC evaluation showed less signs of rejection in eyes transplanted with MHC-II knockout-RPE cells compared to those transplanted with wild-type RPE cells." (highlighted in the main text)

Reviewer #2 (Remarks to the Author):

In this revised manuscript version, the authors have addressed most of the reviewers' concerns. However, the figures appear to be reduced too much in resolution, even when downloading the source files. They look fuzzy when zooming in to the size of a computer screen, especially Figure 1 (only 510x360 pixels), Figure 4 (720x738 pixels), and Figure 6 (644x360 pixels). Apparently Figure 1 was split up without going back to the original resolution files. It is a pity because otherwise they would be great figures.

Response: Please excuse us for keep submitting low resolution images. It seemed like we

had a trouble in TIF conversion, which is now solved. We aimed 600dpi. We hope the resolution is now sufficient.

Also, the order of the supplemental figures does not correspond to their citations in the text.

Response: Thank you for pointing out. We changed the order of Figs 3S and 2S in Supplementary information and arranged the figure numbers accordingly in the main text (highlighted).

Specific comments:

Introduction, p. 3, 2nd paragraph: "unlike retinal cells that express little MHC class II ..." –

replace "little" with "few"

Response: Corrected.

Discussion, p. 11, 1st line: "gave better graft survival" – proposal: "resulted in better graft survival"

Response: Corrected.

There is a typo in the figure legend of Figure S7 (f) "bright filed"

Response: Corrected.

In summary, the text revisions appear to be sufficient, but the figure quality (resolution) needs

to be improved.

Response: Thank you. The resolution of the figures are now improved.

Reviewer #3 (Remarks to the Author):

In the current study, the author introduced interesting findings of PRE transplantation in monkeys, which could be important for preclinical study. Deficiency or knockout of MHCII in RPE cells could avoid immune rejection and improve graft survival in monkeys. My main concern of the paper is that N was relatively low, due to experimental animals, which affected reliability. Overall, the study provided valuable resources for current understanding of PRE transplantation.

Response: As pointed out by the reviewer, the low N was due to experimental animals that became very expensive and limited in supply especially after COVID-19 pandemic, which is a limitation that generally accompanies the studies using monkey models.

The responses of reviewers' comments were in-point, and I believe that the quality of the paper has been improved.

REVIEWERS' COMMENTS:

Reviewer #1 (Remarks to the Author):

The authors addressed the concerns I raised and improved the resolution of the main figures, which is appreciated.

Please also aim for 600dpi resolution for the supplementary figures as it still appears very poor in the current supplementary file.

Thank you for providing the limitations of the study paragraph as part of the manuscript, it does clarify several points that the readership might also have while reading the article.

I suggest to include it at the end of the Discussion instead.

Overall, I am satisfied with the answers from the authors. The quality of the manuscript has considerably improved, so it should be ready for publication.

Reviewer #2 (Remarks to the Author):

The authors have responded to the reviewers' criticism. The figures are now of excellent quality. They have also made all other requested text corrections.

We thank the reviewers for their careful comments. The following are our point-to-point response indicated in blue. We hope this version of our manuscript satisfy the reviewers' concerns.

Reviewer #1 (Remarks to the Author):

The authors addressed the concerns I raised and improved the resolution of the main figures, which is appreciated.

Please also aim for 600dpi resolution for the supplementary figures as it still appears very poor in the current supplementary file.

Thankyou for the comment. Yes, now the resolution of the supplementary figures are increased to 600dpi.

Thank you for providing the limitations of the study paragraph as part of the manuscript, it does clarify several points that the readership might also have while reading the article.

I suggest to include it at the end of the Discussion instead.

Thankyou for the suggestion. We moved this paragraph to the second paragraph of the Discussion section.